# Inactivation of cytidine triphosphate synthase 1 prevents fatal auto-immunity in mice

Claire Soudais [1,2] ✉, Romane Schaus[1], Camille Bachelet [1,2], Norbert Minet [1,2], Sara Mouasni[3], Cécile Garcin[1,2], Caique Lopes Souza[1,2], Pierre David [4], Clara Cousu[5], Hélène Asnagli[6], Andrew Parker[6], Paul Palmquist-Gomes[2,7], Fernando E. Sepulveda [3], Sébastien Storck [5], Sigolène M. Meilhac[2,7], Alain Fischer[1,8,9], Emmanuel Martin [1,9] & Sylvain Latour [1,2] ✉

De novo synthesis of the pyrimidine, cytidine triphosphate (CTP), is crucial for DNA/RNA metabolism and depends on the CTP synthetases, CTPS1 and −2. Partial *CTPS1* deficiency in humans has previously been shown to lead to immunodeficiency, with impaired expansion of T and B cells. Here, we examine the effects of conditional and inducible inactivation of *Ctps1* and/or *Ctps2* on mouse embryonic development and immunity. We report that deletion of *Ctps1*, but not *Ctps2*, is embryonic-lethal. Tissue and cells with high proliferation and renewal rates, such as intestinal epithelium, erythroid and thymic lineages, activated B and T lymphocytes, and memory T cells strongly rely on CTPS1 for their maintenance and growth. However, both CTPS1 and CTPS2 are required for T cell proliferation following TCR stimulation. Deletion of *Ctps1* in T cells or treatment with a CTPS1 inhibitor rescued *Foxp3*-deficient mice from fatal systemic autoimmunity and reduced the severity of experimental autoimmune encephalomyelitis. These findings support that CTPS1 may represent a target for immune suppression.

The cytidine nucleotide triphosphate (CTP) is a key precursor involved in the metabolism of DNA, RNA and phospholipids. CTP is in limiting concentration in cells and originates from two sources: a salvage pathway and a de novo pyrimidine synthesis pathway[1]. The de novo pathway depends on the CTP synthetase (CTPS) enzymatic activity, encoded by two genes: *CTPS1* and *CTPS2*. These genes share 75% homology/identity[2,3], are highly conserved in mouse (90% of homology/identity) and locate on chromosomes 1 and X respectively, both in human and mouse. CTPS1/2 catalyses the ATP-dependent amination of UTP to CTP with ammonia (NH3) transferred from hydrolysed glutamine[4,5]. CTPS activity is involved in cell proliferation and positively correlates with cell proliferation rates[6–10].

We previously reported consequences of CTPS1 deficiency in humans caused by a homozygous frameshift splice mutation (c.1692-1G > C, p.T566Dfs26X) with a founder effect in populations of the Northwest of England[11]. The mutation leads to the skipping of exon 18 resulting in an unstable mutant protein associated with a reduction of more than 80% of total CTPS1 expression[12]. However, the mutation does not impact the CTP synthetase activity, and 10–20% of residual cellular CTPS activity is preserved[12]. All patients developed a combined

[1]Laboratory of Lymphocyte Activation and Susceptibility to EBV infection, Inserm UMR 1163, Institut Imagine, Paris, France. [2]Université de Paris Cité, Paris, France. [3]Laboratory of Molecular Basis of Altered Immune Homeostasis Inserm UMR 1163, Institut Imagine, Paris, France. [4]Transgenesis Platform, Laboratoire d'Expérimentation Animale et Transgenèse (LEAT), Institut Imagine-Structure Fédérative de Recherche Necker INSERM US24/CNRS, UMS3633 Paris, France. [5]Université Paris Cité, CNRS UMR 8253, INSERM U1151, Institut Necker Enfants Malades, F-75015 Paris, France. [6]Step-Pharma, Technoparc du Pays-de-Gex, Saint-Genis-Pouilly, France. [7]Imagine - Institut Pasteur, Unit of Heart Morphogenesis, INSERM UMR1163, F-75015 Paris, France. [8]Collège de France, Paris, France. [9]These authors contributed equally: Alain Fischer, Emmanuel Martin. ✉e-mail: claire.soudais@inserm.fr; sylvain.latour@inserm.fr

immunodeficiency mainly characterised by high susceptibility to viral infections particularly Epstein Barr virus. These patients had no other clinical symptoms suggesting a critical role for CTPS1 in immune cells. Notably, T lymphocytes of patients exhibited an impaired capacity to proliferate while their effector functions like cytotoxicity and cytokine production were preserved except for IL-2 secretion. This selective effect on proliferation of activated T cells is explained by the strong up-regulation of CTPS1 expression in activated T lymphocytes upon T-cell receptor (TCR) activation. Furthermore, B cells also exhibited impaired proliferation when stimulated through the B-cell receptor (BCR). From these studies we concluded that CTPS1 could represent an attractive therapeutic target to selectively inhibit unwanted T/B cell proliferation involved in conditions such as autoimmunity diseases, T cell lymphoma/leukaemia or graft versus host disease (GvHD). Organ transplantation might also represent a good indication for CTPS1 targeting. A selective inhibitor of CTPS1 is currently being evaluated in relapsed and refractory T and B cell lymphoma patients (NCT05463263)[13,14].

To acquire more knowledge on the biology of CTPS1 and CTPS2 in vivo and to better characterise their respective roles at the organism and cellular levels with a stronger focus on the immune system, several mouse models with tissue selective inactivation of *Ctps1* and/or *Ctps2* have been generated and studied here. We also showed that *Ctps1* deletion or pharmaceutical inhibition using a selective small chemical inhibitor of CTPS1 dampened lethal poly-autoimmunity symptoms of the *Scurfy* mice and experimental autoimmune encephalomyelitis further supporting the proposal that targeting *Ctps1* is a potential treatment of autoimmune disorders.

## Results

### Embryonic lethality in mouse caused by *Ctps1* inactivation

To better characterise the role of CTPS1 in vivo, we first developed a mouse model mimicking the human mutation that did not result in any phenotype (Supplementary Fig. 1 and Supplementary Data information). The possible explanation for this discrepancy is that the splicing constraints for *Ctps1/CTPS1* in mouse and human are different and in mouse the mutation did not disturb the splicing[15]. We took advantage of the KO Mouse Project consortium (KOMP, ID:89620) to obtain genetically modified *Ctps1*flox/flox mice allowing conditional inactivation of *Ctps1* (Supplementary Fig. 2a, c). Eventually, *Ctps1*flox/flox animals were crossed with transgenic mouse lines expressing the cre recombinase (*Cre*) under the control of specific promoters (Supplementary Fig. 2d). Crossing with a *CMV-Cre*-transgenic mouse line[16] leading to ubiquitous deletion of *Ctps1* did not produce homozygous *Ctps1*-deficient mice (*Ctps1*ko/ko) at birth, while heterozygous and wild type animals were born, indicating *in utero* lethality when the two alleles of *Ctps1* are deleted (Fig. 1a). To characterise the developmental arrest caused by *Ctps1* gene inactivation, we first recovered fertilised eggs from females and cultured them for 5 days until the blastocyst stage and genotyped them by PCR. The number of *Ctps1*ko/ko blastocysts harvested was not significantly different from the expected Mendelian ratio, indicating that *Ctps1* is not a critical requirement for blastocyst development. We next analysed embryos at different stages of development from embryonic day 6.5 (E6.5) to 9.5 (E9.5) using programmed mating (Fig. 1a–c). No homozygous *Ctps1* deficient embryos were detected by day E9.5 and increasing numbers of embryos with morphological abnormalities were found between day E6.5 and E8.5. Abnormal embryos recovered at E8.5 were arrested at gastrulation stages. Supporting a role of *Ctps1* at this stage, CTPS1 was found to be highly expressed in the E6.5 epiblast before gastrulation (Fig. 1c, d). However, low remaining staining was observed in *Ctps1* deficient embryos probably due to tissue autofluorescence and/or antibody trapping as whole mount mouse embryos is prone to low level of background staining. Thus, our data indicate that *Ctps1* is strictly required for embryo development between day E6.5 and E8.5. This is consistent with the role of cell proliferation during gastrulation as the major

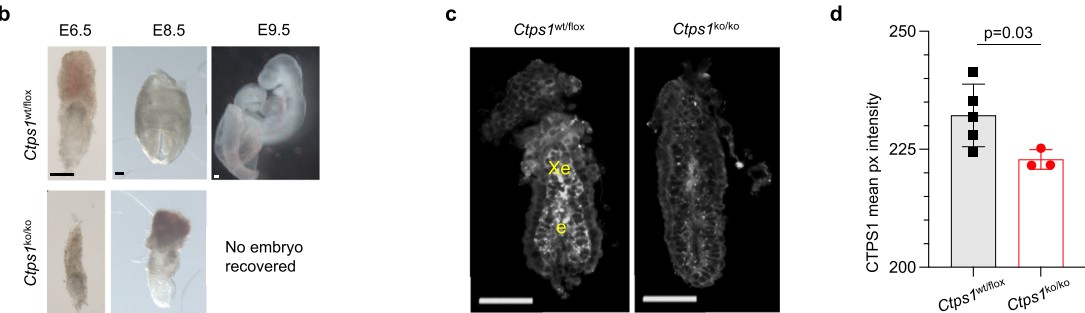

| Developmental stage | Number analysed | *Ctps1*wt/flox | *Ctps1*flox/ko | *CMV-cre ; Ctps1*wt/flox Total (abnormal) | *CMV-cre ; Ctps1*flox/ko Total (abnormal) | Comparison to Mendelian ratio p-value |
|---|---|---|---|---|---|---|
| Blastocyst | 94 | 28 | 19 | 25 | 22 | 5.90E-01 |
| E6.5 | 42 | 9 | 8 | 13 (1) | 12 (4) | 6.55E-01 |
| E8.5 | 31 | 9 | 8 | 6 | 8 (6) | 8.93E-01 |
| E9.5 | 24 | 4 | 12 | 8 | 0 | 3.97E-03 *** |
| P0 | 138 | 43 | 28 | 65 | 0 | 3.54E-14 *** |

**Fig. 1 | CTPS1 is essential for embryonic development. a** Quantification of animals with a given genotype at different stages. *CMV-Cre; Ctps1*wt/ko males were crossed with *Ctps1*flox/flox females. Blastocysts were generated in vitro. Numbers in brackets indicate abnormal embryos. *** significant difference with a chi-square test, indicating embryonic lethality. **b** Brightfield images of *Ctps1*wt/flox (upper panels) and *CMV-Cre; Ctps1*flox/ko (lower panels) at stage E6.5, E8.5 and E9.5 (Scale bars = 100 μm). Images are representatives of 6–13 embryos analysed in three independent experiments. **c** Whole mount immunostaining and 3D imaging of E6.5 embryos with an anti-CTPS1 antibody (grey), which is detected in the epiblast (e) and extra-embryonic ectoderm (Xe) in controls *Ctps1*wt/ko embryos (Scale bars = 100 μm). **d** The mean pixel intensity of the CTPS1 signal from panel (**c**) is significantly decreased in *CMV-Cre; Ctps1*flox/ko mutant embryos. A t-test with Welch's correction is used and data are presented as the mean ± SD of *n* = 5 control and three mutant embryos. Source data are provided in Source Data file.

**Table 1 | List of transgenic mice used in the project associated to their expected expression features**

| Tg-Cre | Compartment/cells affected |
|--------|---------------------------|
| CMV | All cells |
| $ER^{T2}$ | Inducible |
| Vav | Haematopoietic compartment |
| CD4 | All T cells |
| CD8 | Peripheral CD8-T cells |

morphogenetic force that leads to the expansion of the ectoderm, mesoderm and endoderm layers[17,18].

## Tissues with high renewal and proliferation rates are highly dependent on CTPS1

To prevent the embryonic lethality caused by *Ctps1* deletion, we crossed *Ctps1*flox/flox mice to a *Cre-ER*T2 transgenic mouse line enabling inducible deletion of *Ctps1* upon tamoxifen treatment (Table 1)[19]. As expected, in vitro tamoxifen treatment resulted in *Ctps1* deletion and loss of CTPS1 expression in splenic T cells from *Cre-ER*T2; *Ctps1*flox/flox but not from *Ctps1*wt/flox animals (Supplementary Fig. 3a). *Cre-ER*T2; *Ctps1*wt/flox and *Cre-ER*T2; *Ctps1*flox/flox mice were either forced-fed with tamoxifen, to induce *Ctps1* deletion, or with a vehicle (Figs. 2, 3). Deletion of *Ctps1* by tamoxifen treatment in vivo was confirmed at protein level in T lymphocytes and by PCR (Fig. 3d and Supplementary Fig. 3b).

*Ctps1*ko/ko presented a drastic weight loss upon tamoxifen treatment by day 6 leading to their sacrifice at day 14. In contrast to other animal groups, the *Ctps1*ko/ko group showed a marked drop in reticulocytes numbers associated with reduced haemoglobin, red blood cell and lymphocytes numbers, while there were no significant differences in the total number of whole blood cells and neutrophils (Fig. 2a). Considering that the *Ctps1*ko/ko animals were dramatically losing weight, we carefully examined the gut of *Ctps1*ko/ko animals. Haematoxylin-eosin coloration of gut sections revealed aberrant gut structures in *Ctps1*ko/ko animals. The lamina propria was detaching from the epithelium and there was an abnormal cell shedding at the top of the villi. Numerous empty crypts and vacuoles were observed in the tissue (Fig. 2b). In addition, the length of the villi was significantly reduced when compared to all other groups (Fig. 2c). These observations showed that CTPS1 is key for the proliferation and the renewal of the intestinal epithelium of the gut, which is considered as one of the highly proliferative tissues in the whole body[20].

We also analysed Peyer's patches (PP), that host numerous germinal centres (GCs)[21], sites of the most intense activation and proliferation of B cells in the body[22] (Fig. 2d-f). Proportions of T and B cells in PP of *Ctps1*ko/ko were similar to those of *Ctps1*wt/ko (Fig. 2d). We next examined proliferating B cells and T follicular helper cells (T$_{FH}$) from the GCs[23,24]. While CD19+CD95+GL-7+ GC B cells were easily detected in *Ctps1*wt/ko, non-treated *Cre-ER*T2; *Ctps1*flox/flox or *Cre-ER*T2; *Ctps1*wt/flox animals, they were almost absent *Ctps1*ko/ko animals (Fig. 2e and Supplementary Fig. 3c). Likewise, CXCR5+PD-1+ T$_{FH}$ cells from *Ctps1*ko/ko animals were reduced in GCs (Fig. 2f). Thus, these observations show a key role of CTPS1 in the expansion of activated B cells and T$_{FH}$ in GCs. The capacity of mice to mount humoral immune responses against the T-dependent antigen 4-hydroxy-3-nitrophenyl (NP) hapten conjugated to Chicken Gamma Globulin (CGG) was then evaluated. Tamoxifen-treated *Cre-ER*T2; *Ctps1*flox/flox or *Cre-ER*T2; *Ctps1*wt/flox were immunised intraperitoneally (i.p.) with NP-CGG adsorbed on alum (Fig. 2g). At day 14 the relative serum titres of NP-specific IgM and IgG1 antibodies was measured by ELISA. A significant reduction of NP-specific IgM was observed in *Ctps1*ko/ko compared to *Ctps1*wt/ko (Fig. 2h). There was also a tendency towards a reduction NP-specific IgG1 in *Ctps1*ko/ko animals (although it was not statistically different). In parallel, proliferating GC B cells and T follicular helper cells (T$_{FH}$) in the Peyer

patches were analysed. Percentages of CD19+CD95+GL-7+ GC B cells as CXCR5+PD-1+ T$_{FH}$ cells were reduced after immunisation with NP-CGG in *Ctps1*ko/ko mice compared to *Ctps1*wt/ko animals (Fig. 2i). Histology analyses of spleen, after immunisation, confirmed the observations from the Peyer patches showing a reduction of proliferating B cells stained with B220 and PCNA (Fig. 2j). The capacity of mice to mount humoral immune responses against the T-dependent antigen was also tested in immunised C57BL/6 mice treated with the Stp-2 compound a small chemical inhibitor of CTPS1 (see below). A similar reduction of NP-specific IgM titres, a decreased proliferating GC B cells and T follicular helper cells (T$_{FH}$) were observed (Supplementary Fig. 4a–d). Altogether these data indicate that CTPS1 is required for antibody responses.

Thymus of *Ctps1*ko/ko mice were smaller in size, and cellularity was reduced 10-fold compared to control (Fig. 3a). Most cells were CD4-/CD8- double negative, suggesting impaired proliferation capacity of early thymocytes. In parallel, single positive CD4 and CD8 cells tended to accumulate (Fig. 3a). In contrast, spleens of *Ctps1*ko/ko mice display a cellularity and composition of B, NK, dendritic cells, macrophages, and neutrophils subsets similar to those of *Ctps1*flox/flox or *Ctps1*wt/ko animals (Fig. 3b and Supplementary Fig. 3d). However, there was a slight increase of CD8+ T cells in *Ctps1*ko/ko mice (Fig. 3b), with no abnormalities in naive and memory CD4+ and CD8+ T cells (Supplementary Fig. 3e). Because the major immune consequence of CTPS1 deficiency in humans is the impaired capacity of activated T cells to proliferate, expansion of splenic *Ctps1*ko/ko T cells in response to CD3 and CD28 stimulation, was then assessed and found to be profoundly impaired which correlated with decreased CTPS1 expression (Fig. 3c, left panels and d). However, some residual CTPS1 expression was observed in *Ctps1*ko/ko T cells likely due to incomplete *Ctps1* deletion after tamoxifen treatment. Moreover, T cells from vehicle-treated *Ctps1*flox/flox mice failed to proliferate when 4-hydroxy tamoxifen was extemporarily added to the culture medium (Fig. 3c, right panels). Proliferation of sorted splenic B cells from *Ctps1*ko/ko animals upon stimulation with LPS + IL-4 was also significantly reduced (Fig. 3e, left panel), as the proliferation of sorted T cells from *Ctps1*ko/ko animals upon stimulation with anti-CD3 and -CD28 (Fig. 3e, right panel).

Taken together, these data demonstrate that CTPS1 is mandatory for proliferation of cells and tissues with high proliferative and renewal potential such as thymocytes, activated lymphocytes and gut epithelial cells. These results also indicate that CTPS1 deletion even in an adult organism is not compatible with life.

## An essential role of CTPS1 in haematopoiesis

Haematopoiesis, in particular erythropoiesis is a developmental process highly dependent on cell proliferation. To evaluate the role of CTPS1 in haematopoiesis, *Ctps1*flox/flox mice were crossed with the *VAV-Cre* transgenic mouse line (Table 1)[25]. Mendelian distribution of the expected genotypes was found biased. *Ctps1*ko/ko mice were under-represented and did not survive after 6 weeks of age (Fig. 4a). *Ctps1*ko/ko animals were smaller in size when compared to littermate heterozygous animals (Fig. 4b and Supplementary Fig. 5a). A severe anaemia was found in these mice likely indicating an impairment of erythropoiesis (Fig. 4c). The thymus was also markedly reduced in size and cellularity. Double positive cells were reduced in *Ctps1*ko/ko animals while single positive CD4 and CD8 were found in higher percentages, suggesting reduced proliferation of double negative (DN) thymocytes leading to double positive (DP) stage (Fig. 4b, d). However, spleen cellularity was comparable to wild type animals or heterozygotes, but absolute numbers of B and T lymphocytes and NK cells were reduced, while macrophages and granulocytes were not affected (Fig. 4e and Supplementary Fig. 5b, c). Bone marrow (BM) cellularity was markedly reduced in *Ctps1*ko/ko animals. The percentage of Ter119+lin- bone marrow cells[26,27] corresponding to erythrocyte progenitors was strongly decreased, while that of

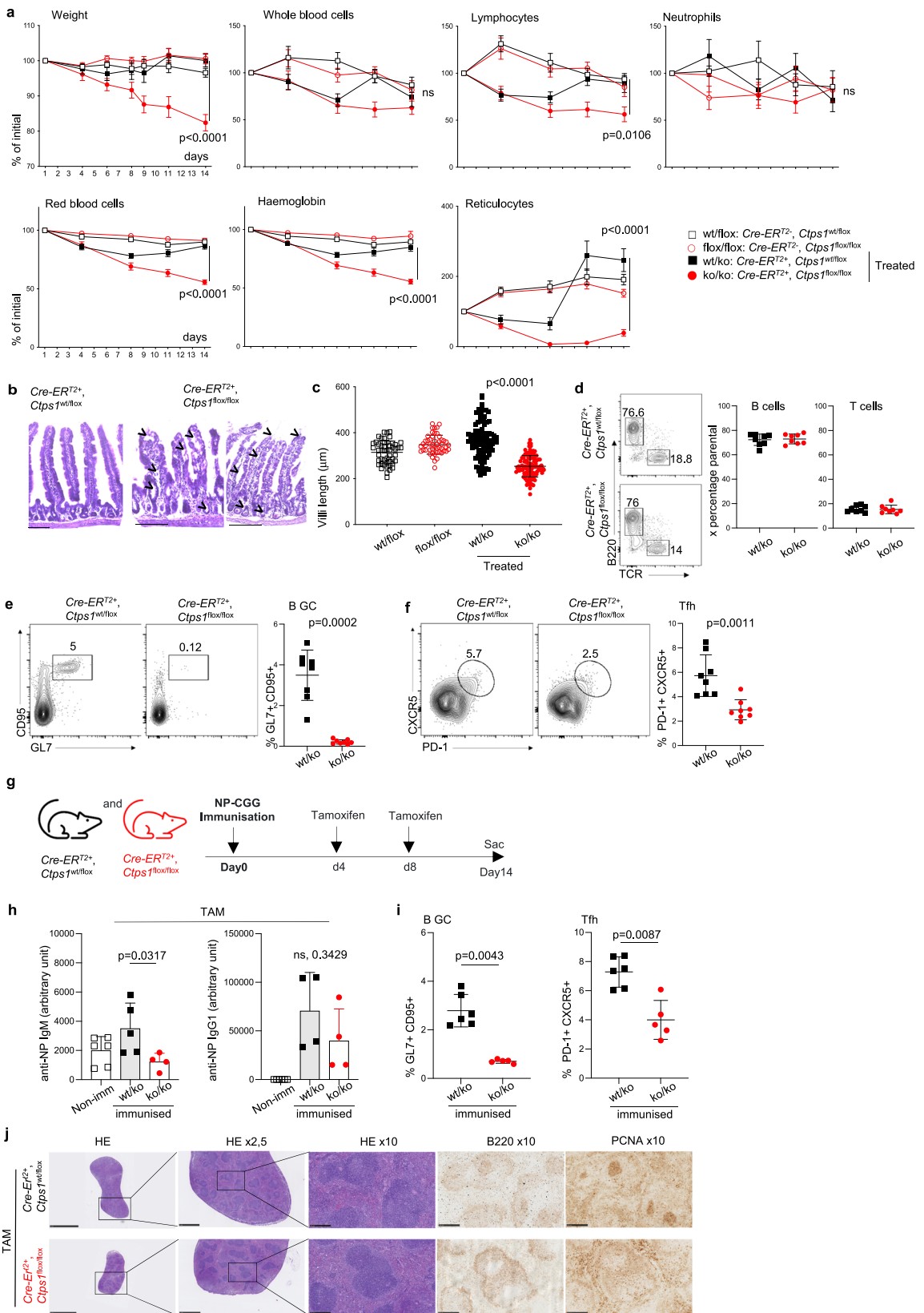

lin⁻Sca1⁺Kit⁺ (LSK⁺) cells which are highly enriched in haematopoietic stem and multi-potent progenitor cells[26–28] was similar to that found in the BM of littermate heterozygous or wild type animals (Fig. 4f and Supplementary Fig. 5d). Absolute numbers of erythroid and hematopoietic progenitors were reduced as bone marrow cellularity of *Ctps1*ko/ko animals was low (Fig. 4f). Therefore, these data

show that *Ctps1* is essential for haematopoiesis of erythroid and lymphoid cells but not for myeloid cells and granulocytes.

## CTPS1 is required for the proliferation of activated T cells

To go deeper into the role of CTPS1 in T lymphocytes activation, *Ctps1*flox/flox mice were crossed to *CD4-Cre-* or *CD8-Cre*-transgenic mouse

**Fig. 2 | Inducible CTPS1 deficiency impairs highly proliferating tissues.**
**a**–**f** Analysis of *Cre-ER^T2*; *Ctps1*^wt/flox and *Cre-ER^T2*; *Ctps1*^flox/flox animals fed with vehicle (*Ctps1*^wt/flox and *Ctps1*^flox/flox or tamoxifen to induce *Ctps1*) gene inactivation (*Ctps1*^wt/ko and *Ctps1*^ko/ko). **a** Weights and blood parameters (whole blood cells, lymphocytes, neutrophils, red blood cells, haemoglobin, and reticulocytes) of animals from day 1 of tamoxifen treatment to day 14. Values are expressed as percentage of the initial population counts at day 0. **b** Representative microscopy images of haematoxylin-eosin coloration of small intestine from tamoxifen treated *Ctps1*^wt/ko and *Ctps1*^ko/ko. Images are representatives of four animals per group. Black arrowhead highlights abnormal features (Scale bars = 100 μm). **c** Intestinal villi length of non-treated *Ctps1*^wt/flox, *Ctps1*^flox/flox, and tamoxifen treated *Ctps1*^wt/ko and *Ctps1*^ko/ko animals measured at day 14. **d**-**f** Analysis of Peyer patches from the intestine. Left panels of representative dot-plots from FACS analysis stained with (**d**) anti-B220 and anti-TCR antibodies, (**e**) anti-CD95 and anti-GL7 or (**f**) anti-CXCR5 and PD-1. Right panels of percentages of B cells and T cells (**d**), germinal centre (GC) B cells (**e**) and T follicular helper cells (Tfh) (**f**) from dot-plots. **g**–**j** *Cre-ER^T2*; *Ctps1*^wt/flox or *Cre-ER^T2*; *Ctps1*^flox/flox were immunised with NP-CGG at day 0 and then treated with tamoxifen at day 4 and 8 to induce *Ctps1* deletion. **g** Experimental design. **h** NP-specific IgM (left), and IgG1 (right) from non-immunised (Non imm.) animals and immunised *Ctps1*^wt/ko and *Ctps1*^ko/ko were quantified by ELISA (arbitrary unit). **i** Percentages of germinal centre (GC) B cells, anti-CD95 and anti-GL7 (left) and T follicular helper cells (Tfh), anti-CXCR5 and PD-1 (right) obtained from dot-plots of *Ctps1*^wt/ko and *Ctps1*^ko/ko. **j** Representative microscopy images of haematoxylin-eosin coloration, B220 and PCNA labelling of spleen from tamoxifen treated *Cre-ER^T2*; *Ctps1*^wt/flox and *Cre-ER^T2*; *Ctps1*^flox/flox. Images are representatives of three animals per group. (Scale bars from left to right: 5 mm, 1 mm and 250 μm). Two-way ANOVA with error bar representing mean ± SEM of *n* = 10 (wt/flox), *n* = 9 (flox/flox), *n* = 17 (wt/ko) and *n* = 14 (ko/ko) animals per group (**a**). Unpaired t-tests two-tailed with error bar representing mean ± SD; measurement of 100 villi length (**c**). Data are presented as the mean ± SD of *n* = 8 (wt/ko) and *n* = 8 (ko/ko) (**d**). **e**–**h** Non-parametric Matt-Whitney two-tailed test were used. Data are presented as the mean ± SD of *n* = 8 animals per group (**e**, **f**). Data are presented as mean with ± SD of *n* = 6 (non-immunised); *n* = 5 (wt/ko) and *n* = 6 (ko/ko) animals per group (**h**) and data are presented as the mean ± SD of *n* = 5 (wt/ko) and *n* = 6 (ko/ko) (**i**). Source data are provided in Source Data file.

lines (Table 1)[29,30]. Mendelian distribution of the different genotypes was normal. No phenotypic differences were detected in the thymus and the spleen of *Ctps1*^ko/ko animals from breeding with *CD4-Cre* and *CD8-Cre*-Tg animals when compared to the littermate control genotypes (Fig. 5a and Supplementary Fig. 6a, d). The expression of the *CD8-Cre* transgene is restricted to peripheral CD8^+ T cells[29], while expression of the *CD4-Cre* transgene starts at the late DN4 stage to single positive (SP) stage leading to the deletion of *Ctps1* in all T cells leaving the thymus[30]. As no phenotypic differences at the different stages of thymic development were detected in the thymus from *CD4-Cre* transgenic mice, expression of the *Cre* transgene could be too late to impact the DN-DP transition during which thymocytes are highly proliferating[31,32]. To confirm this point, we harvested thymuses from *CD4-Cre-Ctps1*^wt/flox and *CD4-Cre-Ctps1*^ko/ko and sorted different thymocyte populations including DN2, DN3, DN4, DP, SP4 and SP8 (Fig. 5b). PCR analysis showed that the Flox allele disappeared in DP, SP4 and SP8 populations consistent with the expression of the *CD4-Cre* transgene after the DN4 stage and accounting for some major phenotypic consequences in thymocytes subsets. Furthermore, no abnormalities were detected in blood, spleen cellularity and cell type composition (B cells, CD4^+ and CD8^+ T cells, NK, dendritic cells, macrophages and neutrophils) of *CD4-Cre-Ctps1*^ko/ko and animals *CD8-Cre-Ctps1*^ko/ko (Supplementary Fig. 6b, c, e). Both CD4 and CD8 enriched splenic *CD4-Cre-Ctps1*^ko/ko T cells showed a marked reduction in proliferation upon CD3/CD28 stimulation, when compared to the proliferation of *CD4-Cre-Ctps1*^wt/flox T cells (Fig. 5c). As expected, activated *CD4-Cre-Ctps1*^ko/ko T cells did not express CTPS1 protein (Fig. 5d). Of note, as described for human T cells[11,12], murine non-activated T cells expressed weak amounts of CTPS1 that are markedly upregulated following TCR activation, while CTPS2 expression remained relatively stable (Fig. 5d). Further analysis of T cell subsets after 3 days of activation, showed that both CD4^+ and CD8^+ T cells from *CD4-Cre-Ctps1*^ko/ko presented a significant reduction of effector memory cells (CD44^+CD62L^-), associated with a relative increase of naive (CD44^-CD62L^+) T cells (Fig. 5e). CD8^+ T cells, but CD4^+ T cells from the spleen of *CD8-Cre-Ctps1*^ko/ko upon activation with anti-CD3/CD28 beads, failed to proliferate leading to an effector memory CD8^+ T cells impairment (Fig. 5f, g). Addition of cytidine to the culture medium during the activation by allowing CTP production through the salvage pathway rescued the proliferation of the CD8^+ T cells (Fig. 5h).

Overall, these data demonstrate the importance of CTPS1 in proliferation of activated T cells and recapitulate observations made with human *CTPS1*-deficient T cells[11,12]. However, both in humans and mice, activated CTPS1-deficient T cells retained a residual capacity to proliferate, indicating that T-cell proliferation is not fully dependent on CTPS1.

## *Ctps2* is not required for mice development

As CTP synthetase activity is dependent on CTPS1 and CTPS2[2], we next wanted to evaluate the role of CTPS2 and its putative redundancy with CTPS1. To that purpose, *Ctps2*^flox/flox animals were obtained by using a EUCOMM construct (Supplementary Fig. 2b, c). *Ctps2*^flox/flox animals were first crossed to *CMV-Cre*-transgenic mice. Viable *Ctps2*^ko/ko animals were obtained. After birth, animals developed with no apparent abnormalities and had normal aging. Absence of the CTPS2 protein expression in *Ctps2*^ko/ko animal was verified in lysates prepared from haematopoietic and non-haematopoietic tissues (Fig. 6a). No phenotypic difference in *Ctps2*^ko/ko mice was detected in blood, thymus, spleen, bone marrow and gut when compared with littermate controls (Supplementary Fig. 7a–c). Proliferation of splenic *Ctps2*^ko/ko T cells in response to CD3/CD28 was also found to be normal and comparable to that of littermate control T cells (Fig. 6b). No abnormalities in differentiation of memory T cells was noticed (Supplementary Fig. 7d). Phenotype and B cells counts were also normal in the spleen of *Ctps2*^ko/ko animals (Supplementary Fig. 7c, e). Proliferating CD19^+CD95^+GL-7^+ B cells in the GCs of Peyer's patches of *Ctps2*^ko/ko animals were found to be present in similar proportions as *Ctps2*^wt/ko or wild-type animals (Supplementary Fig. 7e). In addition, proliferation of *Ctps2*^ko/ko splenic B cells stimulated with LPS plus IL-4 was comparable to that of *Ctps2*^wt/ko B cells (Fig. 6c). These data indicate that the inactivation of *Ctps2* is compatible with life and has no developmental deleterious effects on development in mice. Moreover, *Ctps2* is not mandatory for the proliferation of activated T and B lymphocytes.

## A non-redundant role of CTPS2 in proliferation of activated CTPS1-deficient T cells

However, CTPS2 could partially compensate for the absence of CTPS1 in T lymphocytes explaining the residual proliferation detected in *Ctps1*^ko/ko T cells from *CD4-Cre*; *Ctps1*^flox/flox mice breeding. T cells, in which both *Ctps1* and *Ctps2* were inactivated, were obtained to test this hypothesis. To do so, *CD4-Cre*; *Ctps1*^flox/flox were crossed to *Ctps2*^flox/flox. Like in *Ctps1*^ko/ko animals from *CD4-Cre*; *Ctps1*^flox/flox breeding, cellularity and cell population distribution of blood, thymus and spleen were normal in double CTPS1^ko/ko-CTPS2^ko/ko with the noticeable exception of T lymphocyte counts that were significantly diminished (Fig. 6d and Supplementary Fig. 7f–h). Absence of CTPS1 and/or CTPS2 proteins was checked by western blot. Both CTPS1 and CTPS2 proteins were absent in lysates of enriched activated T cells from double *Ctps1*^ko/ko-*Ctps2*^ko/ko animals. CTPS1 was detectable but not CTPS2 in lysates of activated T cells from *Ctps2*^ko/ko animals, while CTPS2 but not CTPS1 was present in lysates of *Ctps1*^ko/ko activated T cells (Fig. 6e). The deletion of both enzymes had a drastic effect on proliferation of

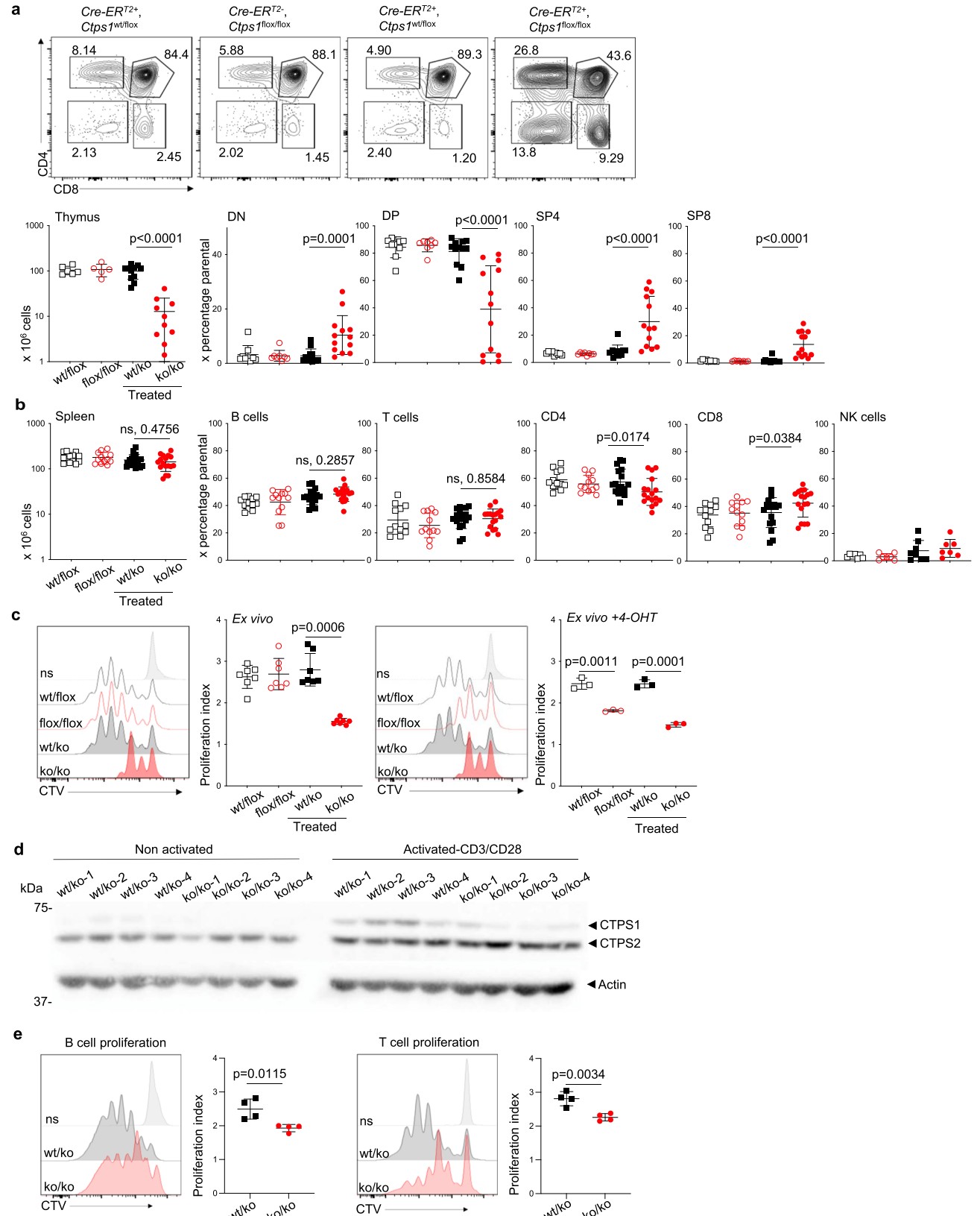

activated CD4[+] and CD8[+] T lymphocytes resulting in an almost complete absence of proliferation (Fig. 6f), while T lymphocytes from *Ctps1*[ko/ko]-*Ctps2*[wt/ko] animals displayed a reduced proliferation, similar to that observed in *CD4-Cre; Ctps1*[flox/flox] T cells. Remarkably some *Ctps1*[ko/ko]-*Ctps2*[ko/ko] CD8[+] T lymphocytes exhibited a residual proliferation that might depend of CTP salvage pathway. Taken together, these results indicate that, although *Ctps1* has an essential role in T cell proliferation that *Ctps2* cannot compensate, the lack of both *Ctps1* and *Ctps2* further reduces T-cell proliferation, implying that CTPS2 has an additional role to CTPS1.

**Fig. 3 | Inducible CTPS1 deficiency impairs thymic development and pro-liferation. a–e** Analysis of thymocytes and splenocytes *Cre-ER^T2^; Ctps1*^wt/flox^ and *Cre-ER^T2^; Ctps1*^flox/flox^ animals fed with vehicle (*Ctps1*^wt/flox^ and *Ctps1*^flox/flox^) or tamoxifen to induce *Ctps1* gene inactivation (*Ctps1*^wt/ko^ and *Ctps1*^ko/ko^). **a** Upper panels corresponding to representative dot-plots from FACS analysis of thymocytes for CD4 and CD8 expression. Lower panels showing total thymus cell counts and percentages of double negative CD4 and CD8 (DN), double positive CD4 and CD8 (DP), single positive CD4 (SP4) and single positive CD8 (SP8) thymocytes from FACS analyses. **b** Spleen cell counts (left panel) and percentages of splenic B, T, CD4, CD8 and NK cells calculated from FACS analyses. **c** Representative proliferation histogram profiles from FACS analysis of enriched spleen T cells labelled with cell trace violet (CTV) and activated ex vivo after tamoxifen treatment, with anti-CD3/CD28 beads plus IL-2 for 3 days. Graph bars (right panel) corresponding to proliferation indexes calculated from histogram profiles (left panels). On the right panels, same as left but 4-hydroxy tamoxifen was extemporary added to the ex vitro culture of

spleen T cells (right panels). **d** Immunoblots for CTPS1, CTPS2 and Actin expression in lysates of non-activated or activated T cells at the day of sacrifice. Four treated *Ctps1*^wt/ko^ and *Ctps1*^ko/ko^ animals were analysed. Blot is the pool of two independent experiments. Molecular weights in kDa on the left. **e** B and T cells were sorted from total spleen of *Ctps1*^wt/ko^ and *Ctps1*^ko/ko^ mice. Both T and B were labelled with CTV. B-cell proliferation was induced by stimulation with LPS and IL-4 for 3 days, while T cell proliferation was induced by anti-CD3/CD28 beads plus IL-2 for 3 days. Graph bars (right panels) corresponding to proliferation indexes calculated from histogram profiles. Non-parametric Matt-Whitney two-tailed test (**a–c** left) and unpaired two-tailed t-tests (**c** right and **e**) were used. Data are presented as the mean ± SD of *n* = 9 (wt/flox), *n* = 8 (flox/flox), *n* = 12 (wt/ko) and *n* = 13 (ko/ko) animals per group (**a**), of *n* = 12 (wt/flox), *n* = 12 (flox/flox), *n* = 17 (wt/ko) and *n* = 17 (ko/ko) animals per group (**b**), of *n* = 7 per group (**c**-left) and of *n* = 4 animals per group (**c** right and **e**). Source data are provided in Source Data file.

## Both genetic deletion and pharmaceutical inhibition of *Ctps1* rescue *Scurfy* mice from lethal autoimmunity

We next tested whether deletion of *Ctps1* could dampen pathological T-cell responses involved in autoimmunity and in inflammatory diseases. We took advantage of the *Scurfy* (Sf) mouse model that mimics the IPEX syndrome (immune dysregulation, poly-endocrinopathy, enteropathy, X-linked) responsible in humans for a severe poly-autoimmunity and inflammatory disorder, caused by hemizygous loss-of-function mutations in the X-linked gene *FOXP3/FoxP3*[33,34]. *FOXP3/FoxP3* gene is critical for the development of CD4^+ regulatory T cells (T_reg)[35,36] which are key to immune tolerance[37]. FOXP3-deficient mice and human males exhibit T cell-dependent inflammation and auto-immunity linked to exacerbated T cell activation and responses among other findings leading to premature death by day 25–35 in mice[38–40] (Fig. 7a). As previously reported[38–40], *Scurfy* (*FoxP3*^Sf/Y^) mice are smaller than their littermate controls and have scaly skin and ears and squinted eyes. Histological evaluation indeed demonstrated thickening of ears and tissue liver disruption with severe inflammation consisting of massive cellular infiltration[38–40] (Fig. 7b–d).

Selective deletion of *Ctps1* in T cells of *Scurfy* mice (afterward designated as *Scurfy-Ctps1*^ko/ko^) was obtained by crossing *CD4-Cre; Ctps1*^flox/flox^ mice to *Scurfy* (*FoxP3*^Sf/Y^). In *Scurfy-Ctps1*^ko/ko^, both CD8^+ and CD4^+ T cells had, as expected, a reduced capacity to expand in response to CD3/CD28 stimulation (Fig. 7e, Supplementary Fig. 8a). Similarly, to *Scurfy* mice, *Scurfy-Ctps1*^ko/ko^ animals exhibit a deficiency in T_reg cells (CD4^+, CD25^+, FoxP3^+) (Fig. 7f), indicating that *Ctps1* deletion has no effect on T_reg differentiation. Importantly, the phenotype of *Scurfy* mice was rescued in *Scurfy-Ctps1*^ko/ko^ (Fig. 7a–d). *Scurfy-Ctps1*^ko/ko^ survived at least up to 80 days and did not present scaly tails and ears while histological analyses did not reveal inflammation and tissue disruption at day 44 (Fig. 7d). Nevertheless, the weight of rescued mice never reached that of control animals (Fig. 7b). Blood parameters were reduced in *Scurfy* animals at day 21, while *Scurfy-Ctps1*^ko/ko^ animals had values comparable to controls, except for monocyte and lymphocyte proportions that were intermediate (Fig. 7g). In the thymus, cellularity and distribution of DP, SP4 and SP8 were profoundly perturbed in *Scurfy* animals and were restored to normal in *Scurfy-Ctps1*^ko/ko^ animals (Fig. 7h), although by day 44, rescued mice developed enlarged spleen while thymuses had a reduced cellularity (Fig. 7c, h). The spleen cellularity was slightly increased in *Scurfy-Ctps1*^ko/ko^ animals, percentages of B and NK cells stayed low, while values of total T cells returned to the normal (Fig. 7i and Supplementary Fig. 8b). However, accumulation of effector memory CD4^+ and CD8^+ T cells seen in the spleen of *Scurfy* mice was significantly reduced in *Scurfy-Ctps1*^ko/ko^ animals, but although not reaching proportions of control mice (Supplementary Fig. 8c). These data thus demonstrate that inactivation of *Ctps1* in T lymphocytes prevents the autoimmune and inflammatory disease in *Scurfy* mice allowing their survival.

Hence, we assumed that targeting CTPS1 with a specific molecule inhibiting its function can suppress the lethal autoimmune symptoms of *Scurfy* mice. A series of small molecules that selectively inhibit human and mouse CTPS1 activity was generated[41]. In contrast to 3-Deazauridine[42], an uridine/UTP analogue, that inhibits both CTPS1 and CTPS2, the Stp-2 compound showed efficient reduction of in vitro T cell proliferation and had the highest selectivity towards murine *Ctps1* among other compounds tested (Supplementary Fig. 8d, e). Stp-2 was injected subcutaneously to *Scurfy* animals from day 10 and every 2 days (Fig. 8a–g). The dose of 30 mg/kg for the Stp-2 compound was chosen based on its solubility and potency. As controls, *Scurfy* males injected with a vehicle died between day 15 to 30. In contrast, a sig-nificant number (50%) of *Scurfy* males treated with Stp-2, survived beyond 50 days (Fig. 8a), had normal mobility and behaviour and did not present severe phenotypical symptoms like scaly tail, and ear and squinted eyes (Fig. 8b). Histological images of ears showed no sign of inflammation or thickening while liver images showed reduced sign of cellular infiltration (Fig. 8d). Nevertheless, their weight did not increase as the one of their treated littermate heterozygous treated sis-ters (Fig. 8c).

Along the treatment of *Scurfy* animals with the compound Stp-2, red blood cell counts and haematocrit returned to normal ranges, while haemoglobin was still slightly reduced as compared to treated littermate heterozygous females (Fig. 8e). The percentage of blood lymphocytes remained low in treated *Scurfy* mice compared to treated females. Blood monocyte and neutrophil proportions remained as high as in non-treated *Scurfy* males (Fig. 8e). Some treated males were sacrificed at day 52–58 as their littermate sisters. Of note, one treated male was kept until 125 days. Heterogeneity in the thymus cellularity was observed among treated males. Indeed, in two mice, thymus cell numbers and thymus subpopulations returned to normal (Fig. 8f), while in four others, these numbers were close to those of non-treated *Scurfy* mice (before they died) (Fig. 7h). Spleen abnormalities in cell population distribution remained similar to that of non-treated *Scurfy* mice (Fig. 8g and Fig. 7i). The results show that targeting Ctps1 with a specific selective small chemical inhibitor improved the lifespan of *Scurfy* animals and several blood parameters returned to normal levels although lymphocyte populations did not recover in contrast to *Scurfy-Ctps1*^ko/ko^.

## Pharmaceutical inhibition of *Ctps1* limits experimental auto-immune encephalomyelitis

The Stp-2 compound was further tested in a more common model of autoimmunity, the induced-experimental autoimmune encephalo-myelitis (EAE). EAE is a T-cell-mediated autoimmune disease char-acterised by lymphocyte infiltration in the central nervous system (CNS) associated with local inflammation, resulting in primary demyelination of axonal tracks, associated impaired axonal

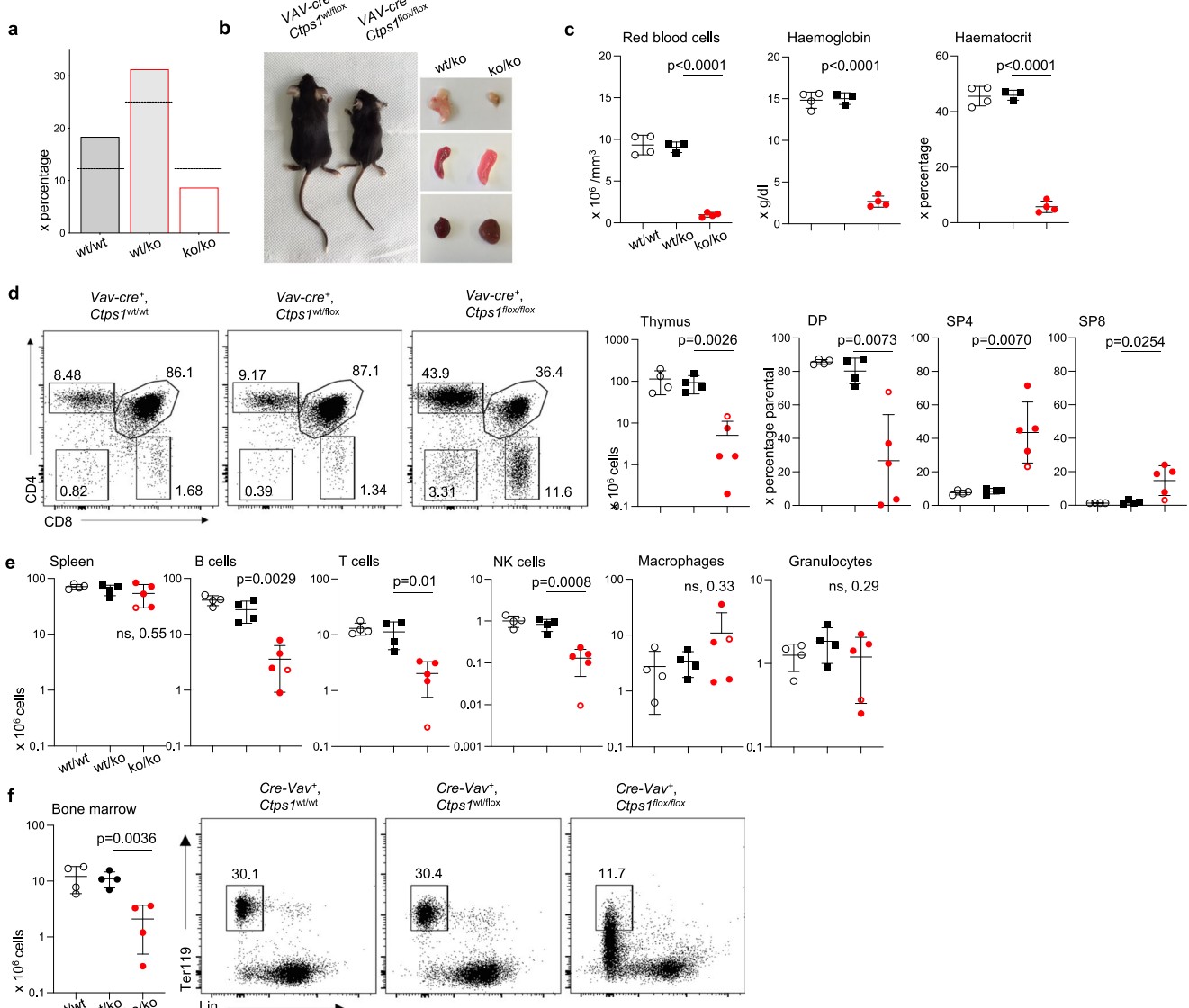

**Fig. 4 | CTPS1 is required for normal development and haematopoiesis. a** Graph bars representing the percentage of animals of each genotype (*Ctps1*wt/wt, *n* = 18 *Ctps1*wt/ko, *n* = 31 and *Ctps1*ko/ko, *n* = 9) obtained from the *VAV-Cre*; *Ctps1*wt/flox breeding (*n* = 100). The horizontal black line represents the percentage of the expected repartition of each genotype. **b** Macroscopic views of *VAV-Cre*; *Ctps1*wt/flox (*Ctps1*wt/ko) and *VAV-Cre*; *Ctps1*flox/flox (*Ctps1*koko) animals and their thymus (upper right), spleen (middle right) and heart (lower right). **c**–**e** Analysis of haematological and immunological parameters at day 15-25. **c** Red blood cell numbers, haemoglobin, and haematocrit. **d** Left panels corresponding to representative dot-plots from FACS analysis of thymocytes for CD4 and CD8 expression. Right panels showing total thymus cell counts and percentages of double negative CD4 and CD8 (DN),

positive CD4 and CD8 (DP), single positive CD4 (SP4) and single positive CD8 (SP8) thymocytes from FACS analyses. **e** Total spleen cell counts (left panel) and proportions of spleen B, T, NK cells, macrophages and granulocytes (right panels) from FACS analysis. **d**, **e** Empty red circle represent a 15-day-old animal. **f** Total counts of bone marrow cells. Representative dot-plots from FACS analysis of red blood cell progenitors (Ter119+) contained in the bone marrows stained with anti-Ter119 and anti-lineages antibodies. **c**, **d**–**f** Unpaired, two-tailed t-tests. Data are presented as the mean ± SD of *n* = 4 (wt/wt), *n* = 3 (wt/ko) and *n* = 4 (ko/ko) animals per group (**c**); of *n* = 4 (wt/wt), *n* = 4 (wt/ko) and *n* = 5 (ko/ko) animals per group (**d**–**e**); of *n* = 4 animals per group (**f**). Source data are provided in Source Data file.

conduction in the CNS, and progressive hind-limb paralysis[43]. C57BL/6 mice were immunised with MOG peptide to induce EAE and then treated with the Stp-2 compound or vehicle every two days post immunisation (Supplementary Fig. 4a). The treatment with Stp-2 allowed a robust reduction of the clinical score along the disease (Fig. 9a). The beneficial effect was stable over time and sustained until day 30. We also measured the general ambulatory ability of CTPS1 inhibitor and vehicle-treated EAE-induced animals compared to non-immunised animals using an open field maze. Confirming the clinical score evaluation, EAE-induced mice treated with Stp-2 showed a good mobility compared to non-immunised mice, while vehicle-treated EAE animals tended to stay at the peripheral of the open field unable to

travel a long distance (Fig. 9b). Mice were sacrificed and their spinal cords were analysed for cellular infiltration and demyelination which are a hallmark of EAE. While vehicle-treated EAE animals presented with extensive lymphocyte infiltration associated with demyelination, Stp2-treated animals were comparable to non-immunised animals lacking signs of infiltration with preserved myelinated areas (Fig. 9c, d). Thus, our data demonstrate that treatment targeting Ctps1 improves EAE clinical signs and controls inflammation and demyelination.

## Discussion
Herein, we show that *Ctps1* but not *Ctps2* is essential for life in mice. In humans, CTPS1 deficiency causes combined immunodeficiency and

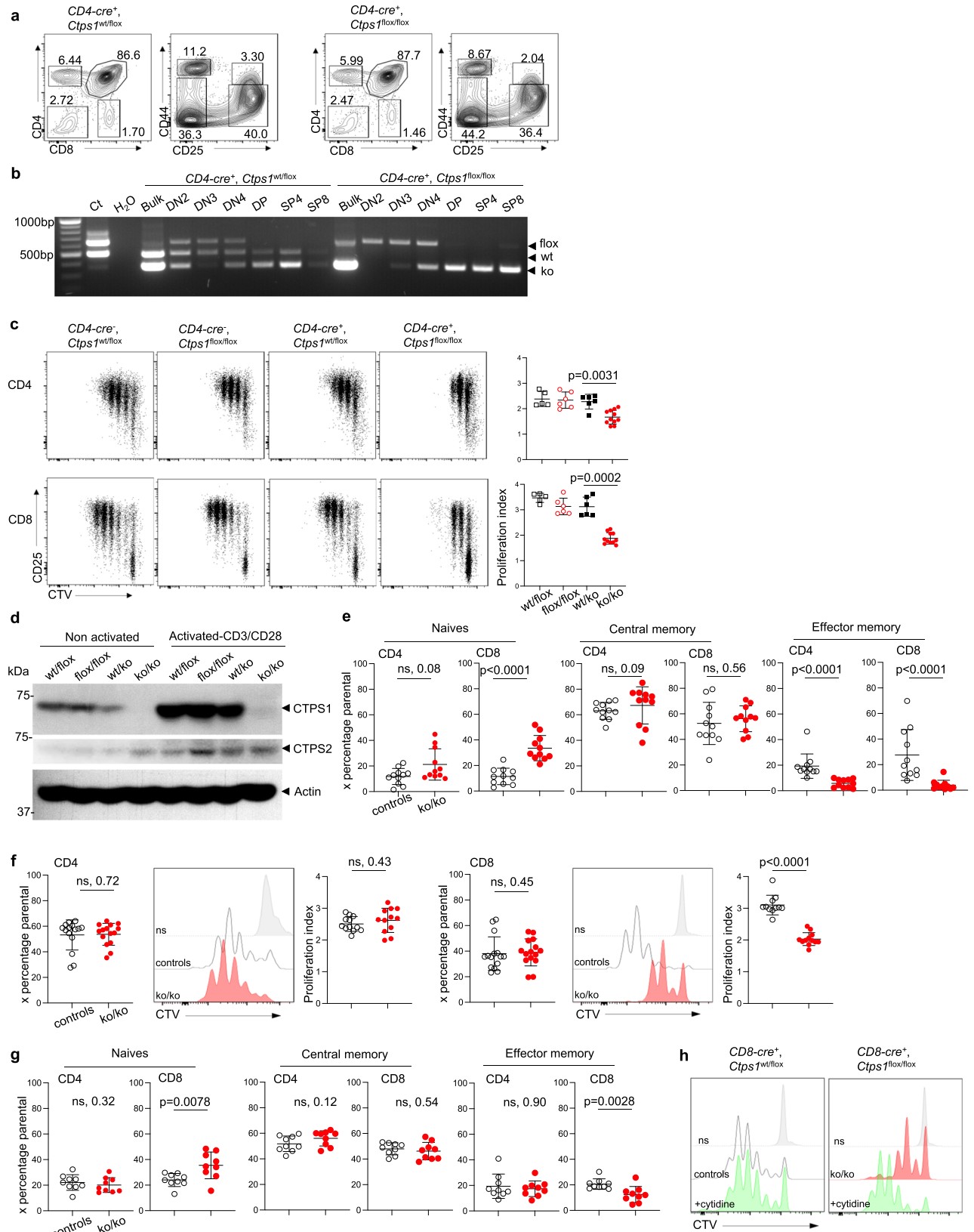

has no developmental effect. However, the mutation causing CTPS1 deficiency in human is hypomorphic and preserved around 10–15% of the CTPS1 expression and activity[12]. This residual CTPS1 activity is in all likelihood sufficient for human normal development further highlighting the key role of CTPS1 in immunity and in particular in lymphocytes. Inactivation of *Ctps2* has no effect on mice development

indicating that *Ctps*1 and *Ctps2* have quite distinct roles. We recently showed that CTPS2 exhibits a lower CTP synthetase activity than CTPS1 that might explain the lesser importance of CTPS2 contribution in cell proliferation[10]. Our results indicate that *Ctps1* is not only mandatory for early development but also for later steps and during all life. Indeed, inducible inactivation of *Ctps1* in normally developed mice results in

**Fig. 5 | CTPS1 inactivation in T lymphocytes leads to impaired expansion and memory T-cell differentiation. a–e** Analyses of *Ctps1*[wt/flox] (*Ctps1*[wt/flox]), *Ctps1*[flox/flox] (*Ctps1*[flox/flox]), *CD4-Cre*; *Ctps1*[wt/flox] (*Ctps1*[wt/ko]) and *CD4-Cre*; *Ctps1*[flox/flox] (*Ctps1*[ko/ko]). **a** Representative dot-plots, of two independent experiments, from FACS analyses of thymic sub-populations stained with CD4 and CD8 double negative CD4 and CD8 (DN) double positive CD4 and CD8 (DP), single positive CD4 (SP4) and single positive CD8 (SP8) cells. DN cells were further labelled with CD44 and CD25 to identify stages of DN differentiation. **b** RT-PCR analyses of the *Ctps1* deletion in total thymus (bulk), sorted cells from DN2, DN3 and DN4 subtypes of DN cells and SP4 and SP8 cells. Corresponding amplified alleles/genotypes are indicated on the right (flox, wt and ko). **c–e** Analysis of splenic enriched T cells activated by anti-CD3/CD28 plus IL-2 for 3 days. **c** Representative proliferation dot plot profiles (left panels) from FACS analysis of T cells. Graph bars (right panels) corresponding to proliferation indexes calculated from histogram profiles on CD4⁺ T cells subset (upper panel), and CD8⁺ T cells subset (lower panel). **d** Immunoblots for CTPS1, CTPS2 and Actin expression in lysates of non-activated or activated T cells. Molecular weights in kDa on the left. **e** Proportions of T cell subsets from FACS analyses in the cultures after activation. Naive T cells (CD44⁻CD62L⁺), central memory (CD44⁺CD62L⁺), effector memory (CD44⁺CD62L⁻) are shown in both CD4⁺ and CD8⁺ T cells. Controls correspond to littermates (*Ctps1*[wt/flox], *Ctps1*[flox/flox] and *Ctps1*[wt/ko]) of

*CD4-Cre*; *Ctps1*[flox/flox] (*Ctps1*[ko/ko]). Blot is representative of four independent experiments. **f–h** Analyses of splenic enriched T cells analyses from *CD8-Cre*; *Ctps1*[wt/flox] (*Ctps1*[wt/ko]) and *CD8-Cre*; *Ctps1*[flox/flox] (*Ctps1*[ko/ko]) activated by anti-CD3/CD28 plus IL-2 for 3 days. **f** Representative proliferation histogram profiles from FACS analysis of enriched *CD8-Cre-Ctps1*[ko/ko] and *Ctps1*[wt/ko] splenic T CD4⁺ (left panels) or CD8⁺ (right panels) cells labelled with cell trace violet (CTV) and activated. White and red histograms correspond to *Ctps1*[wt/ko] and *Ctps1*[ko/ko] respectively. Grey histograms represent unstimulated cells. Graph bars on the right correspond to proliferation indexes calculated from histogram profiles. **g** Proportions of CD4⁺ and CD8⁺ T cell subsets from FACS analyses in the cultures after activation. Naive T cells (CD44⁻CD62L⁺), central memory (CD44⁺CD62L⁺), effector memory (CD44⁺CD62L⁻). Controls correspond to littermates (*Ctps1*[wt/ko]) of *CD4-Cre*; *Ctps1*[flox/flox] (*Ctps1*[ko/ko]). **h** Same as (**f**, right panel) except that CD8⁺ T cells from *Ctps1*[wt/ko] (upper panel) and *Ctps1*[ko/ko] (lower panel) were supplemented (green filled) or not (white or red filled histograms) with cytidine (200 µM). **c**, **e**–**g** Non-parametric Matt-Whitney two-tailed test were used. Data are presented as the mean ± SD of *n* = 5 (wt/flox), *n* = 6 (flox/flox), *n* = 6 (wt/ko) and *n* = 11 (ko/ko) animals per group (**c**), *n* = 11 per group (**e**), *n* = 12 animals per group (**f**) and of *n* = 9 animals per group (**g**). Source data are provided in Source Data file.

rapid health deterioration associated with a rapid weight loss requiring the sacrifice of the animals.

Overall, it seems that the degree of reliance on *Ctps1* in a given tissue (or a cell type/population) positively correlates with its proliferative rate. As such renewal of gut epithelium, production of red blood cells and thymopoiesis are associated with the highest proliferation rate in the organism including T-cell expansion upon antigenic stimulation (see below). Thymic development has been previously shown to be also dependent on the CTP salvage pathway[44,45]. However, our study demonstrates the CTP de novo synthesis pathway is critical for thymic development. Interestingly, in contrast to lymphocytes, myeloid cells appear to be less or not dependent on CTPS activity for their development and maintenance. Thus, in myeloid cells CTP synthesis may be achieved by the salvage pathway rather than by de novo synthesis.

As we previously reported in humans, we observed in the different mouse models that T-cell proliferation upon TCR/antigenic stimulation is highly dependent on CTPS1, as well as GC B cells are. CTPS1 expression in splenic T cells is rapidly up regulated in response to TCR engagement. Interestingly, residual proliferation of activated *Ctps1*-deficient T cells is dependent on CTPS2. We also noticed that few CD8⁺ T cells in absence of both *Ctps1* and *Ctps2* were able to proliferate, while no CD4⁺ T expanded. This could be explained by the different metabolic constraints between CD8⁺ and CD4⁺ T lymphocytes[46]. The CTP salvage pathway could be more active in CD8⁺ T cells than in CD4⁺ T cells.

Inborn errors of purine and pyrimidine metabolism are a diverse group of disorders that present with a wide range of phenotypes including mental retardation, autism, growth retardation, renal stones, and immunodeficiency disorders[47]. Deficiencies in adenosine deaminase (ADA)[48], a housekeeping enzyme of purine metabolism encoded by the *ADA* gene or in the purine nucleoside phosphorylase (*PNP*) deficiency[49], are characterised by recurrent infections, neurologic symptoms, and autoimmune disorders. Both genetic deficiencies lead to a blockade of the purine pathway leading to the intracellular accumulation of deoxynucleosides and deoxynucleotides, which are poisonous for both dividing and non-dividing lymphocytes. Some disorders of the pyrimidine metabolism are also associated with a marked susceptibility to infections such as orotic aciduria (caused by bi-allelic mutations in *UMPS*), and pyrimidine nucleotide depletion syndrome[47]. However, to date *CTPS1*-deficiency is the only genetic defect impairing the pyrimidine pathway with a clinical phenotype restricted to immunodeficiency likely explained by the hypomorphic nature of the *CTPS1* mutation[12].

We provide evidence that inhibiting CTPS1 in T cells significantly attenuates the poly-autoimmune and inflammatory disease of *Scurfy*

mice. Unsurprisingly, inactivation of *Ctps1* appears more efficient than the treatment with the Stp-2 compound. Treatment with Stp-2 allows mice to survive with reduced phenotypic characteristics of the disease. We also showed that the Step-2 compound efficiently reduces the severity of the disease in EAE, a more common model of autoimmunity and inflammation. Therefore, our results provide preliminary evidence to utilise highly selective CTPS1 inhibitors to treat human autoimmune and/or inflammatory conditions that are driven by pathological T-cell responses. Importantly, the study of these different mouse models also informs on possible adverse effects of treatments with CTPS1 inhibitors that might include anaemia, intestine injury, and lymphopenia. However, we did not notice that treatment with Stp-2 had such effects in heterozygous littermate *FoxP3*[wt/sf] females.

Most conventional immunomodulatory agents act by inhibiting activation or reducing proliferation of lymphocytes, notably by targeting purine and pyrimidine biosynthesis. Each of these drugs has its own mechanism of action and is used to manage a variety of autoimmune conditions such as rheumatoid arthritis, psoriasis, systemic lupus erythematosus, systemic sclerosis or organ transplantation. For example, methotrexate inhibit several enzymes responsible for nucleotide synthesis including dihydrofolate reductase, thymidylate synthase, aminoimidazole caboxamide ribonucleotide transformylase (AICART) and, amido-phosphoribosyltransferase[50]. Azathioprine has an antagonist effect on purine metabolism leading to broad inhibition of DNA, RNA, and protein synthesis[51]. Another drug commonly used is the mycophenolate, which is an inhibitor of the inosine-5′-monophosphate dehydrogenase, which result in depletion of guanosine nucleotide preferentially in T and B lymphocytes thus inhibiting their proliferation[52]. Teriflunomide selectively and reversibly inhibits dihydro-orotate dehydrogenase, a key mitochondrial enzyme in the de novo pyrimidine synthesis pathway, leading to a reduction in proliferation of activated T and B lymphocytes without causing cell death[53,54]. Although these drugs, which interfere with nucleotide synthesis, have been shown to be effective, they have more than one mechanism of action and the precise way in which they exert their effects is often unknown. Moreover, their anti-proliferative and cytotoxic effects are in most cases not specific to the immune system explaining their toxicity and side-effects. In our case, we hypothesised based on our acknowledge of CTPS1, that targeting CTPS1 (with selective inhibitors) would be more specific and could lead to less adverse side-effects. Indeed, we widely characterised the role of CTPS1 in proliferation in activated T cells and activated T cells represent one of the highest CTPS1 expressing tissue[11,12]. One possible limitation of the use of CTPS1 inhibitors is to impair the immune response to viral

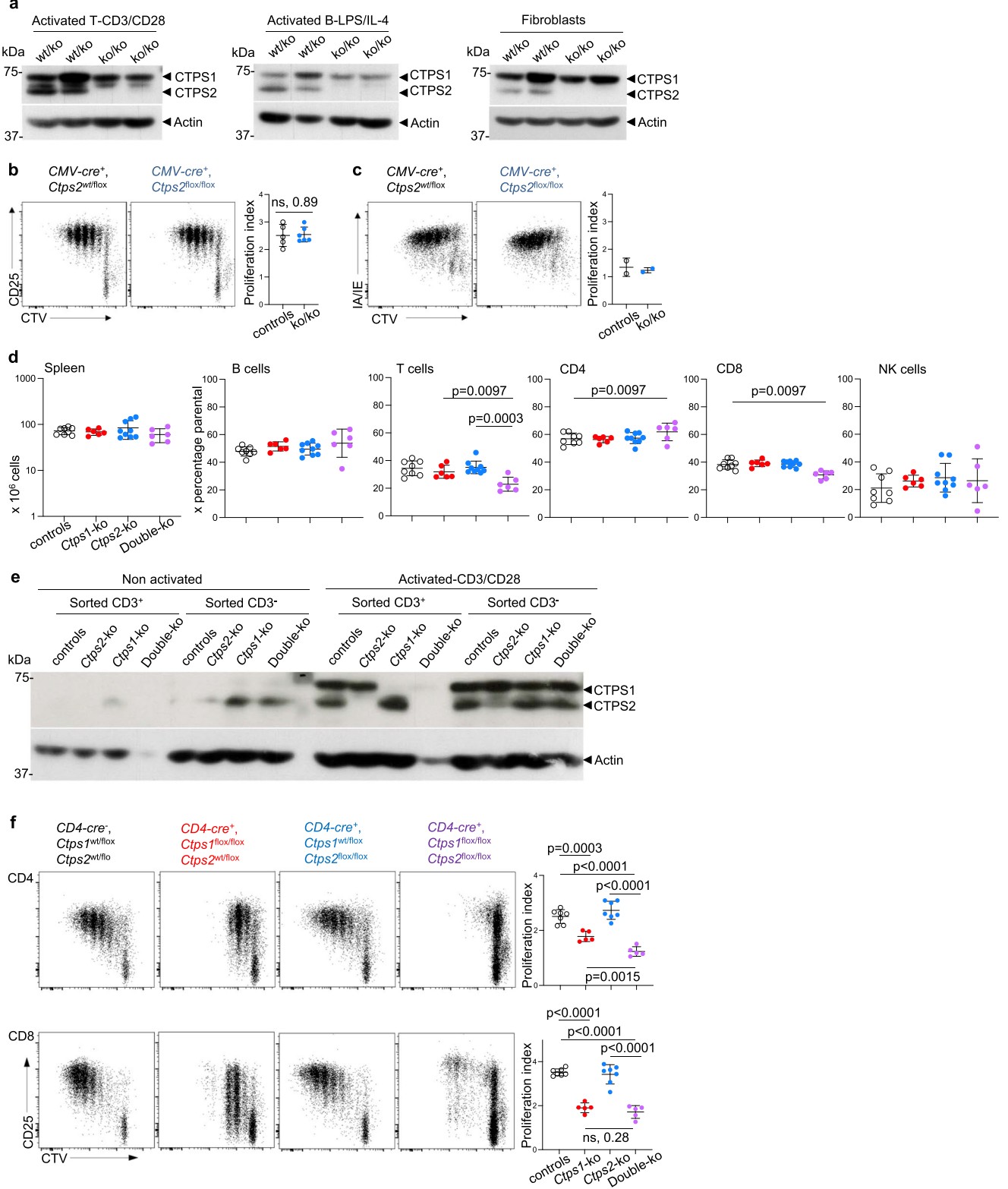

infections including EBV reactivation (as it is observed in *CTPS1*-deficient patients). This unwanted side effect could be resolved by an adjusted dosage of CTPS1 inhibitor (as it is done for other immuno-suppressive drugs).

In conclusion, our study by the establishment and the analyses of these different *Ctps1* deficient mouse models provide important findings on the respective roles of CTPS1 and CTPS2 notably in haematopoiesis and T lymphocyte activation. We also provide several lines of

evidence that CTPS1 represents a promising target for the treatment of autoimmune and inflammatory conditions.

## Methods
### Mouse ethic authorisation
All mouse experiments were performed in accordance with European Union (EU) Directive 2010/63/EU. Animal procedures were approved by the animal committee of the University of Paris Descartes (Paris,

**Fig. 6 | Partial redundancy of CTPS1 and CTPS2 in proliferation of activated T cells. a**–**c** Analyses of T and B lymphocytes of *CMV-Cre; Ctps2*^flox/flox^ (*Ctps2*^ko/ko^) and control littermate (*Ctps2*^wt/flox^, *Ctps2*^flox/flox^ and *Ctps2*^wt/ko^) animals. **a**, **b** Analysis of splenic enriched T cells activated by anti-CD3/CD28 plus IL-2 for 3 days. **a**, **c** Analysis splenic purified B cells activated with LPS plus IL-4 for 3 days. **a** Immunoblots for CTPS1, CTPS2 and Actin expression in lysates of activated T cells (left panel), activated B cells (central panel) and fibroblasts recovered from ear skin (right panel). Molecular weights in kDa on the left. Blots represent two pooled, independent experiments. **b** Representative proliferation dot plot profiles (left panels) from FACS analysis of enriched T cells labelled with cell trace violet (CTV) and stained for CD25, as an activation marker. Graph bars (right panels) corresponding to proliferation indexes calculated from histogram profiles. **c** Representative proliferation dot plot profiles (left panels) from FACS analysis of B cells labelled with cell trace violet (CTV) and stained for IA/IE, as an activation marker. Graph bars (right panels) corresponding to proliferation indexes calculated from histogram profiles. **d**–**f** Analysis of from *CD4-Cre; Ctps1*^wt/flox^x*Ctps2*^wt/flox^ (*Ctps1*^wt/ko^, *Ctps2*^wt/ko^) empty black circle, *CD4-Cre; Ctps1*^flox/flox^x*Ctps2*^wt/flox^ (*Ctps1*^ko/ko^, *Ctps2*^wt/ko^) filled red circle, *CD4-Cre; Ctps1*^wt/flox^x*Ctps2*^flox/flox^ (*Ctps1*^wt/ko^, *Ctps2*^ko/ko^) filled blue circle and *CD4-Cre; Ctps1*^flox/flox^x*Ctps2*^flox/flox^ (*Ctps1*^ko/ko^, *Ctps2*^ko/ko^) filled purple circle. **d** Spleen cell counts (left panel) and percentages of splenic B, T, CD4+, CD8+ and NK cells calculated from FACS analyses. **e** Immunoblots for CTPS1, CTPS2 and Actin protein expression in lysates from non-activated or activated splenic T cells, sorted and non-sorted on CD3+. Molecular weights in kDa on the left. Blot is representative of two independent experiments. **f** Representative proliferation dot plot profiles (right panels) from FACS analysis of CD4+ (upper panels) and CD8+ (lower panels) T cells labelled with cell trace violet (CTV) and stained for CD25 as an activation marker. Graph bars (right panels) corresponding to proliferation indexes calculated from histogram profiles. Non-parametric Matt-Whitney two-tailed test (**b**) and Unpaired t-tests two-tailed (**d**, **f**) were used. Data are presented as the mean ± SD of $n = 5$ animals per group (**b**), of $n = 8$ (controls), $n = 6$ (Ctps1ko), $n = 9$ (Ctps2ko) and $n = 6$ (DKO) animals per group (**d**) and of $n = 7$ (controls), $n = 5$ (Ctps1ko), $n = 7$ (Ctps2ko) and $n = 5$ (DKO) animals per group (**f**). Source data are provided in Source Data file.

France) and the Ministry of Higher Education, Research and Innovation (APAFIS#32465). Animal were housed in individual ventilated cages of group of 5 animals maximum with enrichment, under specific and opportunistic pathogen-free (SOPF) conditions. Animals were fed standard chow diet ad libitum, and kept under ambient temperature (21–22 °C) and 50–60% humidity, with 12–12 h on-off light cycle. Both male and female are used in this study unless specified for specific experiments.

## Mice

To obtain genetically modified mice carrier the same mutation as human, mouse mutated allele was generated through CRISPR/Cas9 mediated HR in C57BL/6J mouse zygotes using CRISPR ribonucleoprotein (RNP) complex direct delivery in pronuclei by electroporation. Briefly a guide targeting intron-exon 18 was engineered by CRISPR/Cas9 technology using CRISPOR web site and the matrix, to mimic the human mutation, was designed (Supplementary Table1). Recombinant Cas9, tracrRNA, crRNA and ssODN were purchased from Integrated DNA technologies. 200 ng/uL tracrRNA:crRNA duplex and 2 μM Cas9 protein were mixed and allowed to form an active rRNP complex for 10 min at room temperature, to which 200 ng/µL ssODN were added in Opti-MEM buffer (31985062, ThermoFisher Scientific). Four weeks-old female mice (Janvier Labs, France) were super-ovulated and mated with C57BL/6J male mice. The next day, zygotes were collected and batches of 50 zygotes were aligned between the electrodes, and repeated pulses of electroporation were delivered by the NEPA21 electroporator, allowing the RNPs/ssODN to enter the zygotes. Surviving zygotes were placed in KSOM media (MR-106-D, Merck Millipore) and cultured overnight at 37 °C and 5%CO2 to reach two-cell stage. Two-cell embryos were transferred into pseudo-pregnant females using standard surgical procedures. F0 founders were genotyped and presence of the mutation was confirmed by sequencing. Founders were backcrossed with C57BL/6J mice to remove potential off-targets and to segregate the mutated allele.

*Ctps1* and *Ctps2* ES cells, purchase from KOMP (ID:89620) and EUCOMM (ID:114201) respectively, were injected into blastocysts to generate founder. Chimeric animals were obtained and bred for germ line transmission. Founders were further backcrossed to C57Bl/6J wild type mice (purchased from Janvier Labs; #000664). These animals were further bred to transgenic mice expressing the flipase transgene (Tg-*FLP*)[55] to delete the KOMP cassette and generate animals with flox alleles (*Ctps1*^flox/flox^). Homozygous *Ctps1*^flox/flox^ animals were then crossed to specific CRE transgenic animals. The following CRE transgenic mouse were used: Tg Flipase JAX stock #012930, *CMV-Cre* JAX stock #006054, *Vav-icre* JAX stock #008610, *CD8-Cre* JAX stock #008766, *CD4-Cre* JAX stock #017336, *Cre-ER*^T2^ JAX stock #008463. The *Scurfy* mice line, stock #004088 was maintained by backcrossing on a B6.129S7-*Rag1*^tm1Mom^/J stock #002216 background, allowing the generation of *FoxP3*^Sf/Sf^ animals. *Rag1*^−/−^ female mice do not develop disease. Crossing of these female mice with wild type C57BL/6J mice resulted in litters in which all the males are *FoxP3*^Sf/Y^. Thus *Rag1*^−/+^ males develop the *Scurfy* phenotype, while heterozygous females do not[56]. All mice were euthanized using cervical dislocation. Age- and sex-matched mice between 6 and 12 weeks of age were used in the experiments.

## Treatment of *Scurfy* mice with Stp-2

Stp-2 is derived compound of Cp277[41] with stronger selectivity against murine CTPS1/CTPS2 (IC50 against mCTPS1 of 15 nM and 586 nM against mCTPS2). A 3 mg/ml solution of Stp-2 compound was prepared in 10% Benzyl alcohol and 90% Castor oil, maximum solubility of Stp-2 being 6 mg/mL. Litters from *Scurfy* breeding were sub-cutaneously injected in the back with 50 µl of Stp-2 (30 mg/kg) starting day ten and every 2 days along the life of the animal.

## Induction of the Cre- *ER*^T2^ transgene

Tamoxifen for in vivo Cre induction or 4-hydroxy tamoxifen (4-OHT) for in vitro experiments, were purchase from Sigma-Aldrich. Tamoxifen was prepared in corn oil to a final concentration of 10 mg/mL. 100 µL of tamoxifen was given by oral gavage using a flexible feeding tube (04 × 040) from ECIMED at day 1 and day 4 of the experiment. 4-OHT was added to the in vitro cell culture at 300 nM.

## Blastocysts

Four to six weeks-old female mice from the *Ctps1*^flox/flox^ line were super-ovulated by intraperitoneal injection of 5 IU PMSG (SYNCRO-PART@ PSMG 600UI, Ceva), followed by 5 IU of hCG (Chorulon 1500 IU, Intervet) at an interval of 46–48 h, and mated with a *CMV-Cre*+; *Ctps1*^wt/ko^ male. The next day, zygotes were collected from the female oviducts and exposed to hyaluronidase (H3884, Sigma-Aldrich) to remove the cumulus cells and then placed in KSOM medium (MR-106-D, Merck) and cultured for 3 days into a CO2 incubator (5% CO2, 37 °C), to reach the blastocyst stage. Blastocysts were recovered in individual tubes and DNA was prepared in 20 µL of DNA lysis buffer. 2 µL were used for the genotyping by Polymerase chain reaction (PCR) amplification.

## Embryo analysis

Two proestrus or oestrus females were mated with one male overnight, plugs were checked in the morning. Females were sacrificed and embryos at E6.5, E8.5 and E9.5 were collected and dissected. Whole embryos were then lysed in DNA lysis buffer and genotyped by PCR. E6.5 embryos were fixed in 4% PFA and stained with anti-CTPS1 antibody. Immunofluorescence on whole mount E6.5 embryos was performed using CUBIC clearing as described[57].

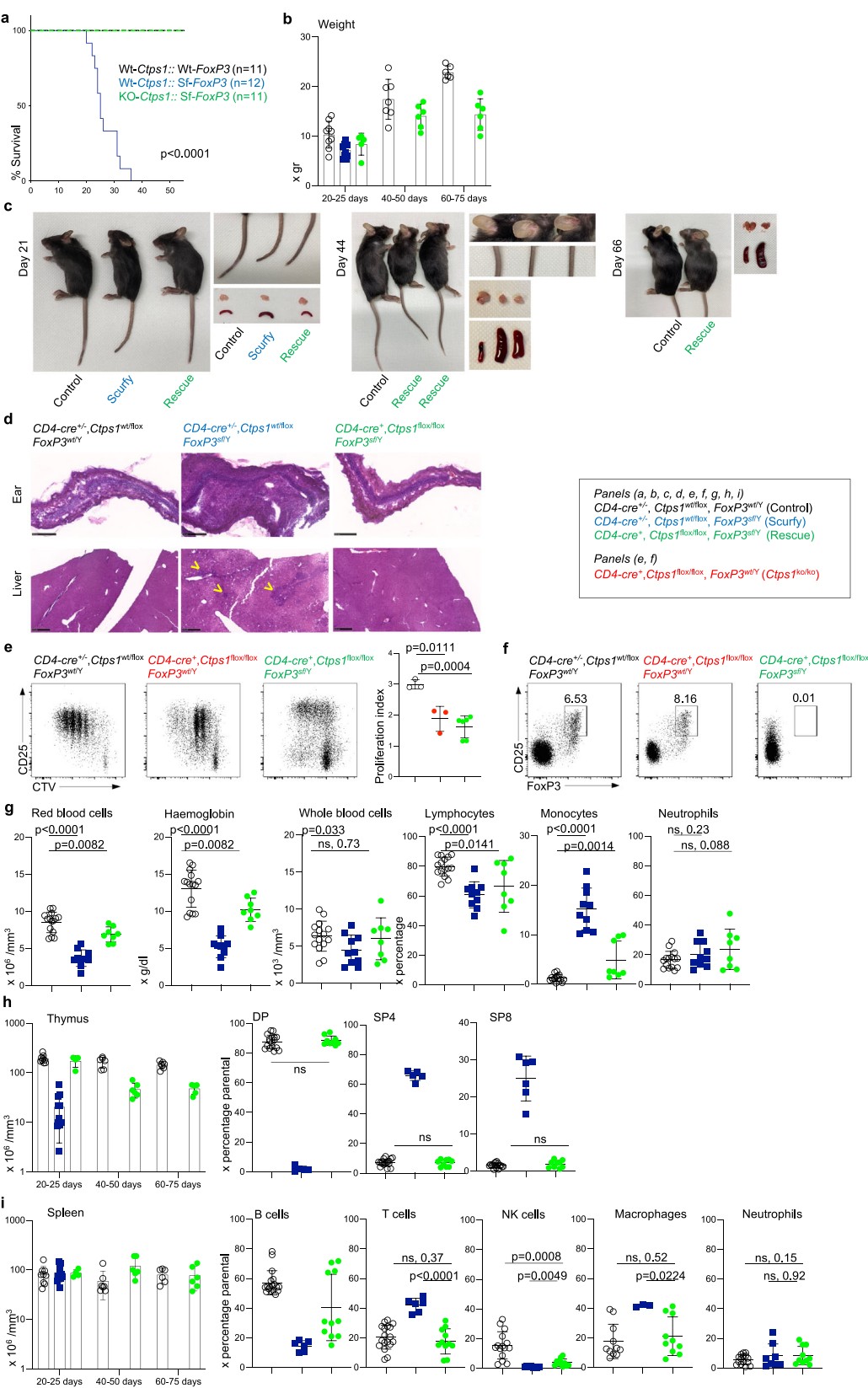

Samples were incubated overnight in the lipid-removing Reagent-1, then 48 h with anti-CTPS1 primary antibody (1/150, ab133743) in TNB blocking buffer (PerkinElmer-FP1020) containing 0.5% triton, and overnight with a Donkey anti-Rabbit Alexa Fluor 488 conjugated secondary antibody (1/500, Invitrogen-A21206) in PBS with 0.01% triton. Samples were finally incubated overnight in Reagent-2 for adjustment of the refractive index and mounted in 0.4% agarose in Reagent-2. Multi-channel 16-bit images were acquired with a Z.1 lightsheet microscope (Zeiss). 3D data was visualised using Imaris software, and CTPS1 mean signal intensity was measured in every

**Fig. 7 | Inactivation of *Ctps1* in T cells in *Scurfy (FoxP3*sf/Y*)* animals relieves autoimmune and inflammatory symptoms. a-i** Analyses of *CD4-Cre*; *Ctps1*wt/flox x*FoxP3*wt/Y (*Ctps1*wt/ko-*FoxP3*wt/Y) *CD4-Cre*; *Ctps1*wt/flox x*FoxP3*sf/Y (*Ctps1*wt/ko-*FoxP3*sf/Y) and *CD4-Cre*; *Ctps1*flox/flox x*FoxP3*sf/Y (*Ctps1*ko/ko-*FoxP3*sf/Y). **a** Kaplan Meier survival curves. **b** Animal weights measured at different days as indicated. **c** Macroscopic views of *Ctps1*wt/ko-*FoxP3*wt/Y, *Ctps1*wt/ko-*FoxP3*sf/Y and *Ctps1*ko/ko-*FoxP3*sf/Y animals and organs. Tail, thymus, and spleen at day 21 (top), day 44 (middle) and day 66 (bottom). Ears also shown at day 44. **d** Representative microscopy images of haematoxylin-eosin coloration of ear and liver at day 21. Infiltration highlighted with yellow arrowheads. Images are representatives of four animals per group. (Scale bars 250 µm). **e** Representative proliferation dot plot profiles (right panels) from FACS analysis of splenic T cells from animals from day 25 labelled with cell trace violet (CTV), stained for CD25 as an activation marker and activated by anti-CD3/CD28 plus IL-2 for 3 days. Graphs (right panel) corresponding to proliferation indexes calculated from histogram profiles. **f** Dot-plots from FACS analysis of FoxP3 intracellular staining on splenic T cells from the same animals as in (**e**). Numbers in the gates correspond to proportions or graph bars of CD4+CD25+FoxP3+. Gates

defined based on staining with control Ig isotypes. **g–i** Analyses of haematological and immunological parameters. **g** Red blood cells, haemoglobin levels, whole blood cells numbers and proportions of lymphocytes, monocytes and neutrophils in blood, analysed from day 18–50. **h** Graph bars showing total thymus cell counts (left panel) at different periods of time as indicated. Proportions (right panels) from FACS analysis of double positive CD4 and CD8 (DP), single positive CD4 (SP4) and single positive CD8 (SP8) thymocytes. **i** Graph bars showing total spleen cell counts (left panel) and proportions of splenic B, T, NK cells, macrophages and neutrophils (right panels) from FACS analysis. A Log-rank (Mantel-Cox) statistical test for curve comparison was used: *n* = 11 (control), *n* = 12 (scurfy) and *n* = 11 (rescue) (**a**). Data are from *n* = 9 (control), *n* = 10 (scurfy) and *n* = 6 (rescue) animals per group (**b**). Unpaired t-tests two-tailed were used (**e**, **g–i**). Data are presented as the mean ± SD of *n* = 3 (control), *n* = 3 (scurfy) and *n* = 6 (rescue) (**e**), of *n* = 15 (control), *n* = 10 (scurfy) and *n* = 8 (rescue) (**g**), of *n* = 17 (control), *n* = 5 (scurfy) and *n* = 9 (rescue) (**h**), and of *n* = 18 (control), *n* = 6 (scurfy) and *n* = 11 (rescue) (**i**). Source data are provided in Source Data file.

---

slide of each stack using FIJI. The mean value of all measurements per embryo is plotted.

### Genotyping
DNA was extracted form ear punch biopsies using the REDExtract-N-Amp™ Tissue PCR kit from Sigma. Polymerase chain reaction (PCR) amplification was then performed with specific primers and analysed on agarose gel after electrophoresis. The list of primers is provided in Supplementary Table 1. For *Scurfy* mice, the *FoxP3* mutation was validated by Sanger sequencing using the BigDye™ Terminator v3.1 Cycle Sequencing Kit (Life Technologies) and a 3500xL Genetic Analyzer (Applied Biosystems) according to the manufacturer's instructions. All collected sequences were analysed using 4peaks software (Version 1.8; A. Griekspoor and T. Groothuis, http://nucleobytes.com/index.php/4peaks). For genotyping of E6.5 mouse embryos, the ectoplacental cone or the whole embryo was lysed. Yolk sac tissue was used for the genotyping of E8.5 and E9.5 embryos. Different stages of thymic T cells differentiation including DN2, DN3, DN4, DP, SP4 and SP8 from were sorted on a FACS Aria. 5000 cells were recovered and directly lysed in DNA lysis buffer and genotyped by PCR.

### Flow cytometry
Cell suspensions were prepared from thymii and spleen by mechanical disruption on cell strainers 70 µm and resuspended on ice in FACS buffer (PBS containing 2% FCS). Prior to staining, all samples were blocked with TruStain FcXt™ PLUS (anti-mouse CD16/32, clone S17011E) from BioLegend in FACS buffer for 15 min on ice. Surface staining was done on ice for 30 min. Surface staining was performed with directly conjugated antibodies (BD Pharmingen, BioLegend and eBioScience) according to standard techniques and analysed on LSR-Fortessa X20 flow cytometer (Becton Dickinson). The following mouse antibodies were conjugated to fluorescein isothiocyanate (FITC), R-phycoerythrin (PE), phycoerythrin-cyanin5 (PE-Cy5), phycoerythrin-cyanin7 (PE-Cy7), Peridinin-chlorophyll-cyanin5.5 (PerCP-Cy5.5), allophycocyanin (APC), allophycocyanin-Cyanin7 (APC-Cy7), alexa-700, Brilliant Violet 421 (BV421), Brilliant Violet 510 (BV510), Brilliant Violet 605 (BV605), Brilliant Violet 650 (BV650), Brilliant Violet 711 (BV711), Brilliant Violet 785 (BV785): anti-TCR (clone H57-597), anti-CD4 (clone RMA4-5), anti-CD8 (clone 53-6.7), anti-CD11b (clone M1/70), anti-CD11c (clone N418), anti-CD19 (clone 6D5), anti-B220 (clone RA3-6B2), anti-CD24 (clone M1/69), anti-CD25 (clone PC61), anti-CD27 (clone LG3A10), anti-CD44 (clone IM7), anti-CD62L (clone MEL-14), anti-CD95 (clone DX2), anti-CD117 (clone 2B8), anti-GR1 (clone RB6-8C5), anti-F4/80 (clone BM8), anti-Ter119 (clone Ter119), anti-Sca1 (clone D7), anti-NKP46 (clone 29A1.4), anti-NK1.1 (clone PK136), anti-IA/IE (clone M5/114.15.2), anti-GL-7 (clone GL-7), anti-CXCR5 (clone L138D7), anti-

PD-1 (clone 29 F.1A12), anti-IgM (clone RMM-1), anti-IgD (clone 11-26c2a).

For intracellular FoxP3 staining, samples were blocked with TruStain FcXt™ PLUS, surfaced labelled then, isotype control-APC (Rat IgG2a, eBR2a) or anti-FoxP3-APC (FJK-16S, eBioScience) antibodies were used with the staining buffer set and protocol from eBioscience. The dilution of antibodies is provided in Supplementary Table 2.

The gating strategies used in all figures and extended data figures are shown in Supplementary Fig. 9.

### T and B-cell proliferation assays
For in vitro T cell activation, T cells were enriched using the Dynabeads Untouched mouse T cells kit from Invitrogen, according to manufacturer. Enriched T cells were further incubated with Cell Trace Violet (CTV) (Biolegend) according to manufacturer and cultured in DMEM +Glutamax supplemented with 10% FCS, penicillin and streptomycin, non-essential amino acids, HEPES and sodium pyruvate and beta mercapto-ethanol (all from Gibco). CTV-labelled enriched T cells were activated with Dynabeads mouse T-activator (anti-CD3/anti-CD28), according to manufacturer plus IL-2 (1000 U/mL) in 48 wells plate at a concentration of 1million/mL for 3 days. 3-deazauridine (3-DU) (Sigma–Aldrich) was used as a non-selective inhibitor and Stp-1 and Stp-2 as specific murine-*Ctps1* inhibitors. 200 µM of cytidine (Sigma–Aldrich) was in some experiments added to the culture to complement proliferation. Sorted B cells, using CD19 beads from Miltenyi. Sorted B cells were incubated with CTV and cultured in DMEM+Glutamax supplemented with 10% FCS, penicillin and strep-tomycin, non-essential amino acids, HEPES and sodium pyruvate and beta mercapto-ethanol (all from Gibco). CTV-labelled B cells were activated with LPS (25 µg/mL; Sigma) plus mIL-4 (25 ng/mL; Peprotec) in 48 wells plate at a concentration of 1million/mL for 3 days. T and B cells proliferation was assessed via CTV dilution after surface staining and analysed by cytometry on LSR-Fortessa X20 flow cytometer (Becton Dickinson), with DIVA software. Data from FACS were processed using FlowJo2 (version 10.7.1, Becton Dickinson & Compagny).

### Biochemical analysis
Lysate from activated and non-activated spleen T cells, activated spleen B cells or fibroblasts were separated on 8% SDS-PAGE gel. After transfer to nitrocellulose membranes, the following primary antibodies were used: rabbit monoclonal anti-CTPS1 EPR8086 (Abcam ab133743) or rabbit polyclonal anti-CTPS2 (C-ter) (Abcam ab190462) (dilution 1/1000) and mouse monoclonal anti-actin (Santa-Cruz-4778) (dilution 1/5000). Membranes were then washed and incubated with anti-mouse (7076S) or anti-rabbit HRP (7074S) conjugated antibodies (dilution 1/10000) from Cell Signalling Technology and Pierce ECL western blotting substrate was used for detection. Uncropped blots are supplied in the Source Data File.

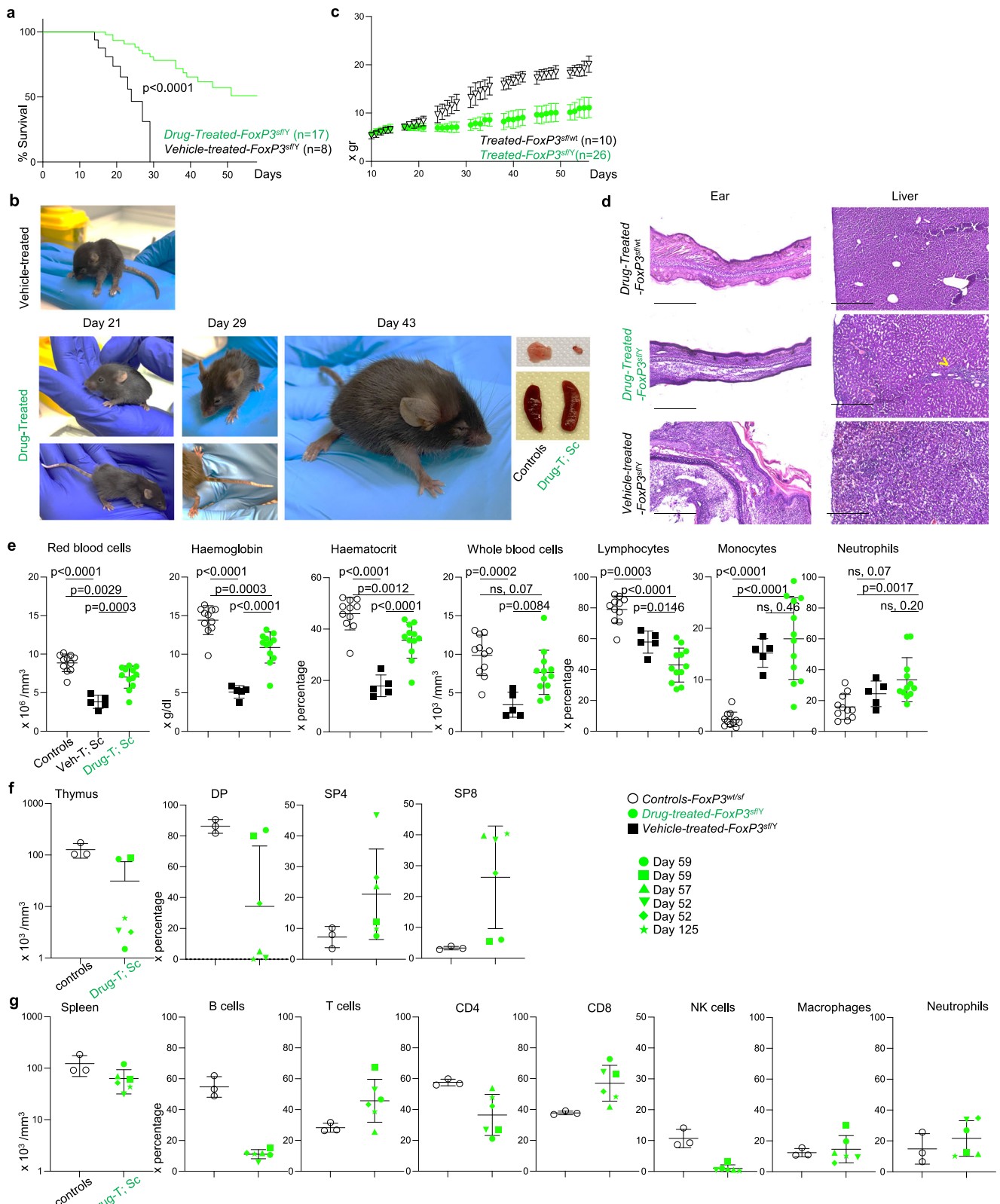

## Histology

Liver and ear were dissected out and fixed in 4% PFA in PBS overnight at 4 °C. The intestine was dissected out, faecal matter was gently removed by using a syringe filled with PBS. The intestine was rolled on itself and placed in a cassette and in fixative solution (4% PFA in PBS) for over-night at 4 °C. The next day organs were washed with cold PBS and left in cold PBS over-night. Two days later, organs were transferred in a 30% sucrose solution for 3 h at 4 °C. Finally, tissues were transferred in cryomolds, quick froze in OCT and kept at −80 °C.

Sections were prepared with an ultramicrotome (RM 2235, Leica). Serial 10-μm sections were cut and collected on glass slides. The

**Fig. 8 | Decreased pathological symptoms and survival of *FoxP3*[sf/Y] animals upon treatment with an inhibitor of CPTPS1. a–d** Analyses of *FoxP3*[sf/Y] vehicle-treated animals (black) or with an inhibitor of *Ctps1* (green) with drug-treated littermate female *FoxP3*[wt/sf] animals as controls (white). **a** Kaplan Meier survival curves. **b** Weights of animals from day 10 and every two-three day along the life of animals of vehicle-treated littermate females *FoxP3*[wt/sf] and males *FoxP3*[sf/Y] treated with the inhibitor of *Ctps1*. **c** Macroscopic view of *FoxP3*[sf/Y] animals at day 21 and *FoxP3*[sf/Y] treated with the Ctps1 inhibitor at day 21, 29 and 43. Thymus, and spleen are shown at day 43. **d** Representative microscopy images of haematoxylin-eosin coloration of ear and liver at day 52. Images are representatives of three animals per group. (Scale bars 500 µm). **e–g** Analyses of haematological and immunological parameters. **e** Red blood cells, haemoglobin levels, haematocrit percentage, whole

blood cells numbers and proportions of lymphocytes, monocytes and neutrophils in blood. **f** Graph bars showing total thymus cell counts (left panel) and proportions (right panels) from FACS analysis of double positive CD4 and CD8 (DP); single positive CD4 (SP4) and single positive CD8 (SP8) thymocytes. **g** Graph bars showing total spleen cell counts (left panel) and proportions of splenic B, T, CD4⁺, CD8⁺, NK cells, macrophages and neutrophils cells (right panels) from FACS analysis. A Log-rank (Mantel-Cox) statistical test for curve comparison was used: $n = 8$ (vehicle-treated), $n = 17$ (drug-treated) (**a**). Data are from $n = 10$ (female, drug-treated), $n = 26$ (male, drug-treated) animals per group (**c**). Unpaired two-tailed $t$ tests were used (**e**, **f**–**g**). Data are presented as the mean ± SD of $n = 11$ (control), $n = 3$ (vehicle-treated), $n = 12$ (drug-treated) (**e**), of $n = 3$ (control) and $n = 6$ (rescue) (**f**–**g**). Source data are provided in Source Data file.

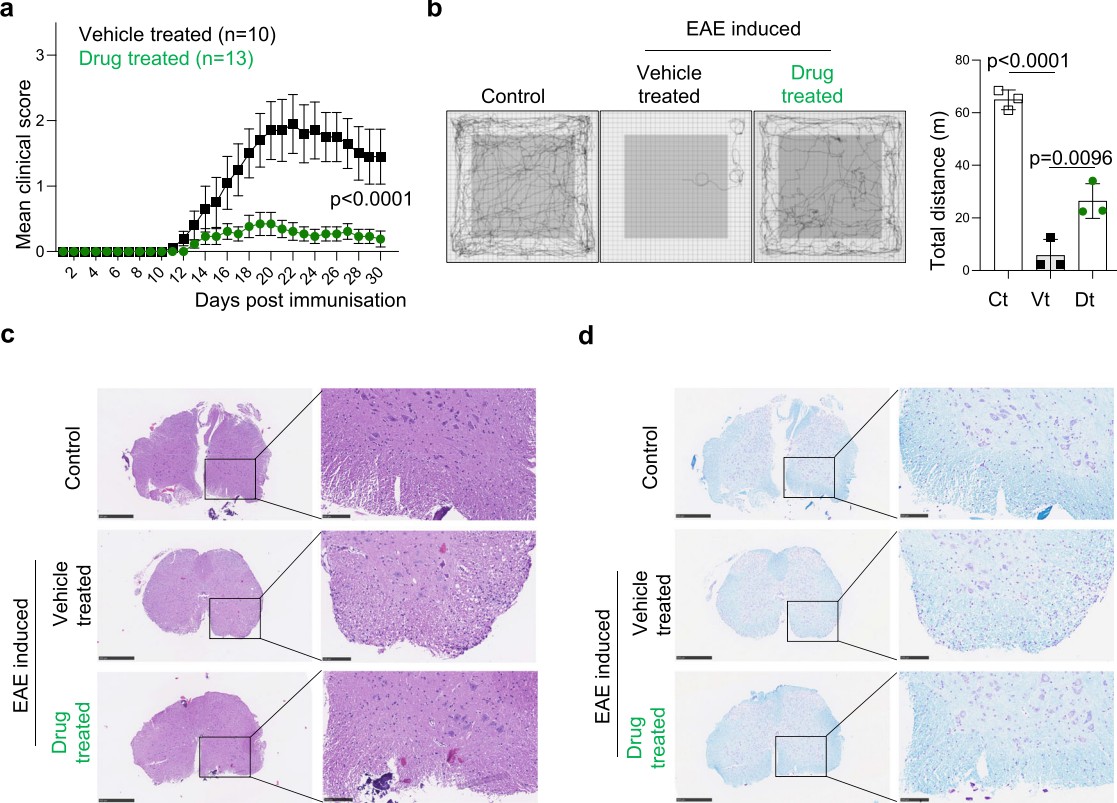

**Fig. 9 | Treatment with an inhibitor of CTPS1 reduces EAE symptoms. a–d** EAE was induced by the immunisation of 12-wk-old C57BL/6 mice with MOG peptide. **a** Clinical score for Stp-2-drug (green) and vehicle (black) treated animals. **b** Representative traces of locomotor activity recorded in an open field cage for 10 min in control mice with no EAE and EAE-induced mouse treated with vehicle (Vt) and in Stp-2 compound (Dt) treated mouse. Quantification of the total distances travelled for 10 min. **c**, **d** Assessment of cellular infiltration by staining with haematoxylin/eosin and luxol blue counterstained with cresyl violet.

Representative haematoxylin-eosin staining (**c**) and Luxol fast blue/cresyl violet counter-staining (**d**) of spinal cord sections of non-EAE and EAE-induced mouse treated with vehicle (Vt) and in Stp-2 compound (Dt) treated mouse. Images are representatives of four animals per group. (Scale bars 500 µm and 100 µm). Two-way ANOVA. Data are presented as mean ± SEM of $n = 10$ (vehicle-treated) and $n = 13$ (drug-treated) animals per group (**a**). One way ANOVA with Tukey's multiple comparisons test $n = 3$ animals per group. Source data are provided in Source Data file.

sections were stained with haematoxylin and eosin (H&E) after drying and photographed with Nanozoomer 2.0 HT microscopy (Hamamatsu).

## Drug testing

Knock-out Jurkat cells for *CTPS1* have been previously reported[7]. KO-Jurkat cells were transduced with a lentiviral vector (pLVX-EF1a-IRES-mCherry (Clontech) with either human-*CTPS1*, human-*CTPS2*, murine-*Ctps1* or murine-*Ctps2*. Transfected cells were seeded at a density of 150000 cells/ml in 96-well plates and treated with the indicated compounds diluted in complete culture medium. 3-deazauridine (3-DU) (Sigma–Aldrich) was used as a non-selective inhibitor and Stp-2 as a specific murine-*Ctps1* inhibitor. At the 24 and 48 h, 10 µL of celltiter-blue (CellTiter-Blue® Cell Viability Assay, Promega) were added to the

culture medium and the cells further incubated at 37 °C for up to 4 h. Absorbance at 560/595 nm was measured and analysed according to the manufacturer's instructions using a Tecan Infinite 200 Pro-plate reader (Tecan Life Sciences).

## NP-CGG immunisation

Immunisations were realised in two animal groups. The first group was *Cre-ER*[T2]; *Ctps1*[wt/flox] and *Cre-ER*[T2]; *Ctps1*[flox/flox] animals in which *Ctps1* deletion was achieved by oral gavage with tamoxifen at day 4 and 8 (Fig. 2g). In the second group, wild type C57BL/6 animals were treated every two days with sub-cutaneous injection in the back with 50 µl of Stp-2 compound (green) or vehicle (black), specifically inhibiting *Ctps1* expression (Supplementary Fig. 4a). Mice were immunised intraperitoneally with 100 µg of NP-CGG (4-hydroxy-3-nitrophenylacetyl-

Chicken Gamma Globulin, Biosearch Technologies) dissolved in PBS and adsorbed on Alhydrogel® aluminium hydroxyde following manufacturer's instructions (Invivogen). Mice were then sacrificed at day 14. Serum was separated after blood clotting by two serial centrifugations for detection of specific IgM and IgG1. Serum from non-immunised C57BL/6 animal was also recovered. Briefly F96 MaxiSorp immunoplates (Nunc) were coated during 1 h at 37 °C with 50 µL per well of a 10 µg.ml$^{-1}$ solution of goat anti-mouse Ig (#1010-01, SouthernBiotech), or of NP-BSA capture antigen (NP23-BSA, Biosearch Technologies) in coating buffer (carbonate-bicarbonate, pH 9.6). Plates were then saturated overnight at 4 °C with 100 µl PBS-1% BSA per well and, incubated for 1 h at RT with 50 µL per well of serum samples serially diluted in PBS. Two duplicate dilution series were performed for each sample. After a last incubation with 100 µl of HRP-conjugated goat anti-mouse IgG1 and IgM (SouthernBiotech) diluted 2 000 times in PBS-1% BSA-0.05% Tween 20, each well was incubated 5–10 min at RT with 50 µl of KPL SureBlue TMB Microwell Peroxidase Substrate (SeraCare) and the reaction was stopped by addition of a 0.6 N $H_2SO_4$ solution. Optical density was measured at 450 nm, background signal measured at 620 nm was subtracted for calculation. For NP-specific Ig titres, a reference pool of six sera taken from C57BL/6 control mice 14 days after immunisation with NP-CGG was used as an internal control on every plate to calculate relative Ig titres. Germinal centres from Peyer patches were analysed by cytofluorimetry as already described (see mat et med in the section Flow cytometry). Spleen were harvested, fixed in 4% PFA and embedded in paraffin. Slides were deparaffinized with Neo-clear (Sigma) and rehydrated. Serial sections (5 µm) were used for haematoxylin-eosin staining (mounted with PERTEX) and immunochemistry. For immune-histochemical assessment, sections were labelled with the following antibodies: biotin conjugated anti-B220 (Invitrogen) and anti-PCNA (Abcam) and a Dako Envision Kit (Dako North America, Inc) was used to reveal the staining. Slides were mounted in Aquatex medium (Merck) and images were taken with Nanozoomer 2.0 HT microscopy (Hamamatsu).

## EAE induction

EAE was induced by subcutaneous immunisation of 12-wk-old C57BL/6 mice at two sites with 100 µg of MOG$_{35-55}$ peptide (MEVG-WYRSPFSRVVHLYRNGK) (Polypeptide Group) emulsified in CFA (Sigma) and supplemented with 6 mg/ml *Mycobacterium tuberculosis* (strain H37RA) (BD). Pertussis toxin (Sigma) (300 ng) was injected intraperitoneal at day 0 and day 2 post-immunization. Clinical score evaluation of symptoms was performed daily according to the standard EAE grading scale. The Stp-2 compound, or vehicle was injected sub-cutaneous every two days starting the day of EAE induction. Disease severity was scored daily on a 5 points scale: 0.5: tip tail is limp; 1: limp tail; 1.5: limp tail and hind leg inhibition; 2: limp tail and weakness of hind legs; 2.5: limp tail and dragging of hind legs; 3: limp tail and complete paralysis of hind legs; 3.5: limp tail and complete paralysis of hind legs plus unable to right itself; 4: complete hind and partial front leg paralysis; 4.5: complete hind and partial front leg paralysis no more movement around; 5: mouse found dead. The general activity of mice was evaluated at day 20 by open field task measurement and recording for 10 min. At day 30 after EAE induction, animals were sacrificed, and lumbar spinal cord sections were harvested fixed in 4% PFA and embedded in paraffin. Deparaffined serial sections (5 µm) were used for eosin-haematoxylin. Adjacent sections were stained with 0.1% Luxol® Fast Blue (Sigma) at 55 °C for 2 h. Sections were dehydrated in 70% alcohol for 15 s and further differentiated in lithium carbonate solution (0.05%) finally counterstained with 0.1% cresyl violet (Sigma) for 5 min. Then, sections were washed, dehydrated in alcohol series, cleared in toluene, mounted in Eukitt medium (Sigma), dried over- night, and photographed with Nanozoomer 2.0 HT microscopy (Hamamatsu).

## Statistics

All graphs and statistical analyses were performed with GraphPad Prism (version 9.4.1; GraphPad Software, Inc, San Diego, CA). Statistical tests included: t-test with Welch's correction, two-way ANOVA, unpaired t-tests, non-parametric Matt-Whitney two-tailed test, log-rank (Mantel-Cox) statistical test for curve comparison and one-way ANOVA with Tukey's multiple-comparison test.

## Reporting summary

Further information on research design is available in the Nature Portfolio Reporting Summary linked to this article.

## Data availability

All data supporting the findings of this study are available within the paper and its Supplementary Information, Source Data file or from the corresponding author upon request (claire.soudais@inserm.fr). Source data are provided with this paper.

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

## Acknowledgements

We thank Geoffrey de Ribains, Tim Bourne and Pascal Georges for their support and discussions over the years. We also acknowledge Virginie Puchois, Corinne Lebreton, Marie Chérier and Julien Diana for their technical and experimental advice. We would like to particularly thank Francina Langa-Vives head of the Mouse Genetics Engineering Center at the Institut Pasteur. We also thank the members of the Latour lab for discussions. This work was supported by grants from Ligue Contre le

Cancer—Equipe Labellisée (France) (to S.L.), Inserm (France), the Agence Nationale de Recherche (ANR, France): ANR-18-CE15-0025-01 (to S.L.), ANR-10-IAHU- 01 (Imagine Institute), ANR-21-CE15-0018-01 (to E.M.) and the European Research Council: Proof of Concept ERC-2015-PoC_Master/ERC-2015-PoC_680465_SAFEIMMUNOSUPPRESS (to A.F., S.L.), INCA PL-BIO2020 (to S.L.). N.M. is a fellowship recipient of the Association Nationale de la Recherche Technologique (ANRT) under agreement with the industrial partner Step-Pharma. P.P.G. is recipient of a Pasteur-Roux-Cantarini fellowship from the Institut Pasteur. S.M.M. is a senior scientist at the INSERM. S.L. is a senior scientist at the Centre National de la Recherche Scientifique (CNRS, France).

## Author contributions

Conceptualisation: C.S., S.L.; Methodology: C.S., S.S., E.M.; Investigation: C.S., R.S., C.B., N.M., S.M., C.G., C.L.S., P.D., C.C., P.P.G., S.S., S.M.M. and E.M.; Resources: H.A., A.P., F.E.S.; Project administration: C.S.; Funding acquisition: A.F., E.M., S.L.; Supervision: S.L.; Visualisation: C.S.; Writing—original draft: C.S., S.L.; Writing—review & editing: C.S., N.M., A.P., F.E.S., S.M.M., A.F., E.M., S.L.

## Competing interests

H.A. was an employee of Step-Pharma SAS. A.P. is a current employee of Step Pharma SAS. N.M. was a fellowship recipient of the Association Nationale de la Recherche Technologique (France) under agreement with Step-Pharma. The remaining authors declare no competing interests.
