## [Peer Review File · Nature Communications]

Inactivation of cytidine triphosphate synthase 1 prevents fatal auto-immunity in mice.Reviewers' Comments:

Reviewer #1:

Remarks to the Author:

The study by Soudais and colleagues generates and analyzes mice with CTPS1 or CTPS2 deficiency, and demonstrates preclinical efficacy of a selective CTPS1-inhibiting small molecule, complementing earlier work from the group identifying hypomorphic CTPS1 mutation as a cause of severe combined immunodeficiency. The results provide definitive information on the critical role of the de novo pyrimidine synthesis as opposed to salvage pathways in particular cell types, notably rapidly dividing T cells, B cells and erythroblasts, and provide a preclinical proof of principle for extension to autoimmune disease of drugs currently being trialled in human lymphoma.

The team first created a mouse with a targeted CTPS1 mutation to recreate the splicing abnormality that diminishes CTPS1 mRNA in humans, but this resulted in a different splicing event in the mice that did not diminish CTPS1.

The team then produced CTPS1 and CTPS2 null mice by Cre-Lox technology. CTPS1 homozygous deficiency from blastocyst stage resulted in early embryonic lethality at E6.5, while acute homozygous CTPS1 deficiency induced in adults resulted in severe deficits in erythropoiesis, thymopoiesis, intestinal villi, and germinal center B cells. Erythropoiesis and lymphopoiesis were also severely impaired by CTPS1 homozygous deficiency selectively in hematopoietic cells created with a Vav-Cre transgene.

Selectively ablating CTPS1 in T cells markedly diminished in vitro T cell proliferation, and this could be corrected by providing exogenous cytidine. Ablating both CTPS1 and CTPS2 almost fully suppressed T cell proliferation.

Finally, the team has used a compound that selectively inhibits CTPS1 to ameliorate lethal autoimmune disease in mice lacking Foxp3+ T cells.

Overall, the experiments are well designed, accurately analysed and interpreted, and the findings are of wide interest, particularly given the preclinical evidence taking a human genetic defect towards a therapeutic.

The one criticism I have is that the discussion is thin. It would be valuable to relate the results here to other genetic deficiencies in purine and pyrimidine synthesis causing immunodeficiency, and to other widely used drugs for treatment of autoimmune diseases such as methotrexate, azathioprine and mycophenolate, which also act by interfering with nucleotide synthesis.

Reviewer #2:

Remarks to the Author:

This is an extremely interesting paper based on the observations of the role of the two genes in patients leading to an immune deficiency.

However there appears to be a number of gaps in this current paper.

In terms of assessment of the immune system of these mice.

Q1 The assessment of the effect of absence or blockade of CPCTS1 on antibody generation. Given the effect shown on TFH cells it is surprising that antibody levels and responses are not measured in these mice.

Q2 Also given the comments on germinal centres there is no histology of the spleen to show GCs in the various mice or any assessment of antibody response with blockade.

The model used to test CPCTS1 role in autoimmune disease is scurfy mice an extremely rare autoimmune disease due to absence of Tregs with foxp3 mutations.

Q3 Surely a model of either spontaneous autoimmune disease such as one of the lupus models or an induced model such as EAE would more accurately reflect its potential role as a therapeutic option.

Q4 Similarly GVHD models are routinely used but have not been tested here despite the observation that this was a potential place for use.

Q5 The risks given the susceptibility to EBV in patients with only partial blockade is of concern given the risks of EBV driven lymphoma and autoimmune disease and these limitations to clinical use are not discussed. Similarly the role of EBV in several autoimmune diseases warrants discussion.

Q6 There is no discussion of solid organ transplantation, given this is an area of T cell immunosuppression and also one where EBV disease is important warrants some discussion.

Q7 The GI issues seen in the mice are not adequately explained and would warrant some transfer studies of normal cell subsets to evaluate if they are protective.

Q8: Is it possible to do some single cell analysis of the immune systems of the mice to better define the populations affected?

Reviewer #3:

Remarks to the Author:

In this work, the authors used mouse models with tissue selective inactivation of CTPS1 and CTPS2, coding for CTP synthetases of the pyrimidine biosynthetic pathway, to elucidate their roles in cell proliferation. They observed that highly proliferative tissues, including activated B and T lymphocytes, and memory T cells, rely heavily on CTPS1. Previous work on the combined immunodeficiency caused by mutation in CTPS1 in humans, led the authors to test the effect of CTPS1 deletion or inhibition on the lethal poly-autoimmunity of Scurfy mice. Their results suggest that CTPS1 should be explored as a potential drug target for treatment of autoimmune and inflammatory diseases. This manuscript represents an extensive amount of work that brings us closer to understanding the role of CTP synthetases in the immune response.

Some minor revisions are required.

INTRODUCTION:

Define GVHD.

What is the percent identity between human CTPS1, CTPS2, and between the human and mouse enzymes?

Although the authors provide references from the 1970s and 1980s on the role of CTPS in cell proliferation and cancer, additional, more current citations on the state of this research should be provided.

RESULTS:

Provide a possible explanation for why the knock-in of the mutation mimicking the human CTPS1 mutation did not affect CTPS1 expression in the mouse model.

A general comment that applies to many figures: the text in the labeling of figures looks out of focus, and is hard to read when in superscripts, see for example Fig. 2 b – e, Fig. 4 b, Fig. 4 d, Fig. 4 f, Fig. 5 a – c, etc.

Fig. 1a: “Nb” is not a standard abbreviation for “number”.

Fig 1d. Why isn't the Ctps1 antibody signal decreased more in the Ctps1 ko/ko embryos? There appears to be substantial signal remaining compared to the Ctps1 wt/flox mice.

Care should be taken to use the same notation in the text and in the figures. The text of the Results, p.5, refers to ERT2, in Supplementary Fig. 2d, the notation is ERT₂, and in Supplementary Fig. 3 the notation is ERT2.

Fig. 2a. How many animals are represented?

Fig 2b: In the legend, refer to “black arrowheads” instead of “black sign”.

Fig. 2 d-f: Correct the legend. The left panels are dot-plots, and the right panels are percentages of B and T cells.

It would be useful to provide the reader with the table shown in Supplementary Fig. 2d in the manuscript, instead of in the supplementary material.

Fig. 3d: In the immunoblots, right panel (Activated-CD3/CD28), lanes wt/ko-1, and lane wt/ko-4 look like they have the same intensities of CTPS1 signal as the lane with ko/ko-1. What is the half-life of CTPS1? Would one expect to see residual protein, left over from before the tamoxifen treatment?

Fig 7d: In the legend, refer to “yellow arrowheads” instead of “yellow signs”.

DISCUSSION:

In the discussion the authors should comment that the pyrimidine biosynthetic pathway has long been considered a target for the treatment of autoimmune disease, although the focus has been on dihydroorotate dehydrogenase (see for example Scherer et al. 2023, PMID: 36797499, doi: 10.1038/s41590-023-01436-x).

Barbara H. Zimmermann

Point-by-point response to Reviewers

Point-by-point response to Reviewer #1 (expert in rare autoimmune/lymphoproliferative disorders):

The study by Soudais and colleagues generates and analyzes mice with CTPS1 or CTPS2 deficiency, and demonstrates preclinical efficacy of a selective CTPS1-inhibiting small molecule, complementing earlier work from the group identifying hypomorphic CTPS1 mutation as a cause of severe combined immunodeficiency. The results provide definitive information on the critical role of the de novo pyrimidine synthesis as opposed to salvage pathways in particular cell types, notably rapidly dividing T cells, B cells and erythroblasts, and provide a preclinical proof of principle for extension to autoimmune disease of drugs currently being trialled in human lymphoma.

The team first created a mouse with a targeted CTPS1 mutation to recreate the splicing abnormality that diminishes CTPS1 mRNA in humans, but this resulted in a different splicing event in the mice that did not diminish CTPS1.

The team then produced CTPS1 and CTPS2 null mice by Cre-Lox technology. CTPS1 homozygous deficiency from blastocyst stage resulted in early embryonic lethality at E6.5, while acute homozygous CTPS1 deficiency induced in adults resulted in severe deficits in erythropoiesis, thymopoiesis, intestinal villi, and germinal center B cells. Erythropoiesis and lymphopoiesis were also severely impaired by CTPS1 homozygous deficiency selectively in hematopoietic cells created with a Vav-Cre transgene.

Selectively ablating CTPS1 in T cells markedly diminished in vitro T cell proliferation, and this could be corrected by providing exogenous cytidine. Ablating both CTPS1 and CTPS2 almost fully suppressed T cell proliferation.

Finally, the team has used a compound that selectively inhibits CTPS1 to ameliorate lethal autoimmune disease in mice lacking Foxp3⁺ T cells.

Overall, the experiments are well designed, accurately analysed and interpreted, and the findings are of wide interest, particularly given the preclinical evidence taking a human genetic defect towards a therapeutic.

The one criticism I have is that the discussion is thin. It would be valuable to relate the results here to other genetic deficiencies in purine and pyrimidine synthesis causing immunodeficiency, and to other widely used drugs for treatment of autoimmune diseases such as methotrexate, azathioprine and mycophenolate, which also act by interfering with nucleotide synthesis.

We have addressed these points. For the first part of the comment, we have now tried to relate our results to other genetic deficiencies in purine and pyrimidine synthesis leading to immunodeficiencies. We have modified our discussion accordingly by proposing to add the following paragraph, lines 436-448, page 14:

“Inborn errors of purine and pyrimidine metabolism are a diverse group of disorders that present with a wide range of phenotypes including mental retardation, autism, growth retardation, renal stones, and immunodeficiency disorders⁴⁷. Deficiencies in Adenosine deaminase (ADA)⁴⁸, a housekeeping enzyme of purine metabolism encoded by the ADA gene or in the purine nucleoside phosphorylase (PNP) deficiency⁴⁹, are characterized by recurrent infections, neurologic symptoms, and autoimmune disorders. Both genetic deficiencies lead to a blockade of the purine pathway leading to the intracellular accumulation of deoxynucleosides and deoxynucleotides, which are poisonous for both dividing and non-dividing lymphocytes. Some disorders of the pyrimidine metabolism are also associated with a marked susceptibility to infections such as orotic aciduria (caused by bi-allelic mutations in *UMPS*), and pyrimidine nucleotide depletion syndrome⁴⁷. However, to date CTPS1-deficiency is the only genetic defect impairing the pyrimidine pathway with a clinical phenotype restricted to immunodeficiency likely explained by the hypomorphic nature of the CTPS1 mutation¹².”

In the second part of his comment, raising the need to relate our observations to other widely used drugs for treatment of autoimmune diseases, Reviewer #1 echo the comments of reviewer #3 (“*In the discussion the authors should comment that the pyrimidine biosynthetic pathway has long been considered a target for the treatment of autoimmune disease, although the focus has been on dihydroorotate dehydrogenase (see for example Scherer et al. 2023, PMID: 36797499, doi: 10.1038/s41590-023-01436-x*”).

We thus address both comments by adding the following paragraph, **lines 460-484, page 15**:
“Most conventional immunomodulatory agents act by inhibiting activation or reducing proliferation of lymphocytes, notably by targeting purine and pyrimidine biosynthesis. Each of these drugs has its own mechanism of action and is used to manage a variety of autoimmune conditions such as rheumatoid arthritis, psoriasis, systemic lupus erythematosus, systemic sclerosis or organ transplantation. For example, methotrexate inhibit several enzymes responsible for nucleotide synthesis including dihydrofolate reductase, thymidylate synthase, aminoimidazole carboxamide ribonucleotide transformylase (AICART) and, amido-phosphoribosyltransferase⁵⁰. Azathioprine has an antagonist effect on purine metabolism leading to broad inhibition of DNA, RNA, and protein synthesis⁵¹. Another drug commonly used is the mycophenolate, which is an inhibitor of the inosine-5'-monophosphate dehydrogenase, which result in depletion of guanosine nucleotide preferentially in T and B lymphocytes thus inhibiting their proliferation⁵². Teriflunomide selectively and reversibly inhibits dihydro-orotate dehydrogenase, a key mitochondrial enzyme in the de novo pyrimidine synthesis pathway, leading to a reduction in proliferation of activated T and B lymphocytes without causing cell death^{53, 54}. Although these drugs, which interfere with nucleotide synthesis, have been shown to be effective, they have more than one mechanism of action and the precise way in which they exert their effects is often unknown. Moreover, their anti-proliferative and cytotoxic effects are in most cases not specific to the immune system explaining their toxicity and side-effects. In our case, we hypothesized based on our acknowledge of CTPS1, that targeting CTPS1 (with selective inhibitors) would be more specific and could lead to less adverse side-effects. Indeed, we widely characterized the role of CTPS1 in proliferation in activated T cells and activated T cells represent one of the highest CTPS1 expressing tissue ^{11, 12}. One possible limitation of the use of CTPS1 inhibitors is to impair the immune response to viral infections

including EBV reactivation (as it is observed in CTPS1-deficient patients). This unwanted side effect could be resolved by an adjusted dosage of CTPS1 inhibitor (as it is done for other immunosuppressive drugs).”

Point-by-point response to Reviewer #2 (expert in immunological tolerance and autoimmunity):

This is an extremely interesting paper based on the observations of the role of the two genes in patients leading to an immune deficiency.

However there appears to be a number of gaps in this current paper.

In terms of assessment of the immune system of these mice.

Q1 The assessment of the effect of absence or blockade of CPCTS1 on antibody generation. Given the effect shown on TFH cells it is surprising that antibody levels and responses are not measured in these mice.

We thank reviewer #2 for this interesting comment. We have now addressed this point by conducting immunization experiments and analysed antibody levels following immunization. We assessed the capacity of mice to mount humoral immune responses against the T-dependent antigen 4-hydroxy-3-nitrophenyl (NP) hapten conjugated to Chicken Gamma Globulin (CGG). We have used either tamoxifen induced CTPS1 deleted animals or wild type animals treated with the small chemical inhibitor of CTPS1 (the experimental plan is shown on Supplementary Fig. 4a). We injected mice intraperitoneally (i.p.) with NP-CGG adsorbed on alum, serum was collected at day 14 and, the relative serum titres of NP-specific IgM and IgG1 antibodies was measured by ELISA. We show a significant reduction of the IgM primary response in both animals in which CTPS1 has been deleted or inhibited when compared to controls groups (Fig. 2g Supplementary Fig. 4b). In our experiment, the primary IgG1 responses were not statistically different in absence or blockade of CTPS1 (although there is a tendency to diminution). The absence of difference seen for IgG1 could be explained by the kinetics of secretion of IgM versus IgG1 during the extrafollicular phase of the antibody response which likely contributes to most of the antigen-specific antibody secretion in this experimental setting. Altogether our data show that CTPS1 is mandatory for a functional antibody response.

Q2 Also given the comments on germinal centres there is no histology of the spleen to show GCs in the various mice or any assessment of antibody response with blockade. We took advantage of our immunisation experiments to analyse this response. We have analysed by flow cytometry the proliferating B cells and T follicular helper cells (T_{FH}) from the GC in the Peyer patches after NP-CGG immunisation. The percentage of $CD19^+CD95^+GL-7^+$ GC B cells as $CXCR5^+PD-1^+T_{FH}$ cells are reduced after immunisation with NP-CGG in absence or blockade of CTPS1 (Fig. 2h and Supplementary Fig. 4c). This data confirms that the intense proliferation in germinal centre affect both B and T_{FH} cells that are deficient for CTPS1 or in which CTPS1 is blocked. In parallel we performed histology analyses of spleen after immunisation. Confirming our observations from the Peyer patches, we observed both in animals in which CTPS1 has been deleted or inhibited a reduction of proliferating B cells

(stained with B220 and PCNA). Altogether these data indicate that CTPS1 is required for the proliferation of B cells and T cells in germinal centre (Fig. 2i and Supplementary Fig. 4d).

These data (for Q1 and Q2) are now shown in Figure 2 panels g, h and i and Supplementary Fig 4. of the revised manuscript and are now discussed in the result section and the material and methods.

Results section, lines 172-189, page 6-7:

“The capacity of mice to mount humoral immune responses against the T-dependent antigen 4-hydroxy-3-nitrophenyl (NP) hapten conjugated to Chicken Gamma Globulin (CGG) was then evaluated (Fig. 2g). Tamoxifen-treated *Cre-ER^{T2}; Ctps1^{flox/flox}* or *Cre-ER^{T2}; Ctps1^{wt/flox}* were immunized intraperitoneally (i.p.) with NP-CGG adsorbed on alum. At day 14 the relative serum titres of NP-specific IgM and IgG1 antibodies was measured by ELISA. A significant reduction of NP-specific IgM was observed in *Ctps1^{ko/ko}* compared to *Ctps1^{wt/ko}*. There was also a tendency towards a reduction NP-specific IgG1 in *Ctps1^{ko/ko}* animals (although it was not statistically different). In parallel, proliferating GC B cells and T follicular helper cells (T_{FH}) in the Peyer patches were analysed. Percentages of $CD19^+CD95^+GL-7^+$ GC B cells as $CXCR5^+PD-1^+$ T_{FH} cells were reduced after immunisation with NP-CGG in *Ctps1^{ko/ko}* mice compared to *Ctps1^{wt/ko}* animals (Fig. 2h). Histology analyses of spleen, after immunisation, confirmed the observations from the Peyer patches showing a reduction of proliferating B cells stained with B220 and PCNA (Fig. 2i). The capacity of mice to mount humoral immune responses against the T-dependent was also tested in immunised C57BL/6 mice treated with the St-2 compound a small chemical inhibitor of CTPS1 (see below). A similar reduction of NP-specific IgM titres, a decreased proliferating GC B cells and T follicular helper cells (T_{FH}) were observed (Supplementary Fig. 4). Altogether these data indicate that CTPS1 is required for antibody responses.”

Material and methods, lines 667-697, page 21-22:

“NP-CGG immunisation

Immunisations were realized in two animal groups. The first group was *Cre-ER^{T2}; Ctps1^{wt/flox}* and *Cre-ER^{T2}; Ctps1^{flox/flox}* animals in which *Ctps1* deletion was achieved by oral gavage with tamoxifen at day 4 and 8. In the second group, wild type C57BL/6 animals were treated every two days with sub-cutaneous injection in the back with 50 μ l of Stp-2 compound or vehicle, specifically inhibiting *Ctps1* expression (Supplementary Fig. 4a). Mice were immunized intra-peritoneally with 100 μ g of NP-CGG (4-hydroxy-3-nitrophenylacetyl-Chicken Gamma Globulin, Biosearch Technologies) dissolved in PBS and adsorbed on Alhydrogel[®] aluminium hydroxyde following manufacturer's instructions (Invivogen). Mice were then sacrificed at day 14. Serum was separated after blood clotting by two serial centrifugations for detection of specific IgM and IgG1. Briefly F96 MaxiSorp immunoplates (Nunc) were coated during 1h at 37°C with 50 μ l per well of a 10 μ g.ml⁻¹ solution of goat anti-mouse Ig (#1010-01, SouthernBiotech), or of NP-BSA capture antigen (NP23-BSA, Biosearch Technologies) in coating buffer (carbonate-bicarbonate, pH 9.6). Plates were then saturated overnight at 4°C with 100 μ l PBS-1% BSA per well and, incubated for 1h at RT with 50 μ l per well of serum samples serially diluted in PBS. Two duplicate dilution series were performed for each sample. After a last incubation with 100 μ l of HRP-conjugated goat anti-mouse IgG1 and IgM (SouthernBiotech) diluted 2 000 times in PBS-1% BSA-0.05% Tween 20, each well was incubated 5-10 min at RT with 50 μ l of KPL SureBlue TMB Microwell Peroxidase Substrate (SeraCare) and the reaction was stopped by addition of a 0.6 N H₂SO₄

solution. Optical density was measured at 450 nm, background signal measured at 620 nm was subtracted for calculation. For NP-specific Ig titres, a reference pool of six sera taken from C57BL/6 control mice 14 days after immunization with NP-CGG was used as an internal control on every plate to calculate relative Ig titres. Germinal centres from Peyer patches were analysed by cytofluorimetry as already described (see mat et med in the section Flow cytometry). Spleen were harvested, fixed in 4% PFA and embedded in paraffin. Slides were deparaffinized with Neo-clear (Sigma) and rehydrated. Serial sections (5 µm) were used for haematoxylin-eosin staining (mounted with PERTEX) and immunohistochemistry. For immunohistochemical assessment, sections were labelled with the following antibodies: biotin conjugated anti-B220 (Invitrogen) and anti-PCNA (Abcam) and a Dako Envision Kit (Dako North America, Inc) was used to reveal the staining. Slides were mounted in Aquatex medium (Merck) and images were taken with Nanozoomer 2.0 HT microscopy (Hamamatsu).

The model used to test CPCTS1 role in autoimmune disease is scurfy mice an extremely rare autoimmune disease due to absence of Tregs with *foxp3* mutations. Q3 Surely a model of either spontaneous autoimmune disease such as one of the lupus models or an induced model such as EAE would more accurately reflect its potential role as a therapeutic option.

We thank reviewer #2 for this relevant suggestion and think this experiment will highly benefit to the paper. To that extend we have immunized C57BL/6 mice with MOG peptide to induce EAE disease. C57BL/6 mice were treated with the small chemical inhibiting CTPS1 or vehicle every two days post immunisation. The drug treatment allows a robust reduction of the clinical score along the disease. We then measured the general ambulatory ability of drug- and vehicle-treated EAE-induced animals compared to non-immunised (non EAE) animals using an open field maze. As confirm by the clinical score evaluation, mice treated with the CTPS1 inhibitor were able to move around the open field, while vehicle-treated animals tended to stay at the peripheral of the open field unable to travel a long distance. We sacrifice animals and spinal cords were stained with haematoxylin/eosin and luxol blue to assess cellular infiltration which is a hallmark of EAE and demyelination. While vehicle-treated EAE animals presented with extensive infiltration and demyelination, animals treated with the CTPS1 inhibitor were comparable to non-immunized (non-EAE) animals lacking signs of infiltration with preserved myelinated areas. Our data indicate that treatment targeting CTPS1 improves EAE clinical signs and controls inflammation and demyelination.

These data have been added in an additional new Figure 9 and are now discussed in the result section, discussion section and the material and methods.

Results section: **lines 384-403, page 13:**

“Pharmaceutical inhibition of *Ctps1* limits experimental autoimmune encephalomyelitis

The Stp-2 compound was further tested in a more common model of autoimmunity, the induced-experimental autoimmune encephalomyelitis (EAE). EAE is a T-cell-mediated autoimmune disease characterized by lymphocyte infiltration in the central nervous system (CNS) associated with local inflammation, resulting in primary demyelination of axonal tracks, associated impaired axonal conduction in the CNS, and progressive hind-limb paralysis⁴³. C57BL/6 mice were immunised with MOG peptide to induce EAE and then treated with the Stp-2 compound or vehicle every two days post immunisation. The treatment with Stp-2 allowed a robust reduction of the clinical score along the disease (Fig. 9a). The beneficial effect was stable over time and sustained until day 30. We also measured

the general ambulatory ability of CTPS1 inhibitor and vehicle-treated EAE-induced animals compared to non-immunised animals using an open field maze. Confirming the clinical score evaluation, EAE-induced mice treated with Stp-2 showed a good mobility compared to non-immunized mice, while vehicle-treated EAE animals tended to stay at the peripheral of the open field unable to travel a long distance (Fig. 9b). Mice were sacrificed and their spinal cords were analysed for cellular infiltration and demyelination which are a hallmark of EAE. While vehicle-treated EAE animals presented with extensive lymphocyte infiltration associated with demyelination, Stp2-treated animals were comparable to non-immunized animals lacking signs of infiltration with preserved myelinated areas (Fig. 9c and d). Thus, our data demonstrate that treatment targeting Ctps1 improves EAE clinical signs and controls inflammation and demyelination.”

Discussion section, lines 452-454, page 15:

“We also showed that the Step-2 compound efficiently reduces the severity of the disease in EAE, a more common model of autoimmunity and inflammation.”

Material and methods, lines 699-720, page 22-23:

“EAE induction

EAE was induced by subcutaneous immunization of 12-wk-old C57BL/6 mice at two sites with 100 µg of MOG₃₅₋₅₅ peptide (MEVGWYRSPFSRVVHLYRNGK) (Polypeptide Group) emulsified in CFA (Sigma) and supplemented with 6 mg/ml *Mycobacterium tuberculosis* (strain H37RA) (BD). Pertussis toxin (Sigma) (300 ng) was injected intraperitoneal at day 0 and day 2 post-immunization. Clinical score evaluation of symptoms was performed daily according to the standard EAE grading scale. The Stp-2 compound, or vehicle was injected sub-cutaneous every two days starting the day of EAE induction. Disease severity was scored daily on a 5 points scale: 0.5: tip tail is limp; 1: limp tail; 1.5: limp tail and hind leg inhibition; 2: limp tail and weakness of hind legs; 2.5: limp tail and dragging of hind legs; 3: limp tail and complete paralysis of hind legs; 3.5: limp tail and complete paralysis of hind legs plus unable to right itself; 4: complete hind and partial front leg paralysis; 4.5: complete hind and partial front leg paralysis no more movement around; 5: mouse found dead. The general activity of mice was evaluated at day 20 by open field task measurement and recording for 10 minutes. At day 30 after EAE induction, animals were sacrificed, and lumbar spinal cord sections were harvested fixed in 4% PFA and embedded in paraffin. Deparaffined serial sections (5 µm) were used for eosin-haematoxylin. Adjacent sections were stained with 0.1% Luxol® Fast Blue (Sigma) at 55 °C for 2 h. Sections were dehydrated in 70% alcohol for 15 seconds and further differentiated in lithium carbonate solution (0.05%) finally counterstained with 0.1% cresyl violet (Sigma) for 5 minutes. Then, sections were washed, dehydrated in alcohol series, cleared in toluene, mounted in Eukitt medium (Sigma), dried over- night, and photographed with Nanozoomer 2.0 HT microscopy (Hamamatsu).”

Q4 Similarly GVHD models are routinely used but have not been tested here despite the observation that this was a potential place for use.

We agree with reviewer #2 that GvHD is an interesting model to test the effect of deletion of CTPS1 and/or inhibition of CTPS1. However, for a question of time, it was not possible to set up a GVHD model in a reasonable time frame to test the effect of CTPS1 blockage. We have preferred to focus on autoimmunity and set up the EAE autoimmunity model that is more

relevant to what was already tested/shown in our manuscript with in the fatal autoimmunity in Foxp3-deficient mice.

Q5 The risks give the susceptibility to EBV in patients with only partial blockade is of concern given the risks of EBV driven lymphoma and autoimmune disease and these limitations to clinical use are not discussed. Similarly, the role of EBV in several autoimmune diseases warrants discussion.

We agree that pathogenicity of autoimmune diseases such as systemic lupus erythematosus or rheumatoid arthritis are complex disorders involving genetic background and environmental factors, including viruses including EBV.

We have addressed the limitations to clinical use of CTPS1 inhibitors **lines 481-484, page 15** in the discussion:

“One possible limitation of the use of CTPS1 inhibitors is to impair the immune response to viral infections including EBV reactivation (as it is observed in CTPS1-deficient patients). This unwanted side effect could be resolved by an adjusted dosage of CTPS1 inhibitor (as it is done for other immunosuppressive drugs).”

Q6 There is no discussion of solid organ transplantation, given this is an area of T cell immunosuppression and also one where EBV disease is important warrants some discussion. We have now addressed these points raised by reviewer#2 in the introduction and in the discussion:

Introduction: **lines 101-102, page 3:**

“Organ transplantation might also represent a good indication for CTPS1 targeting.”

Discussion: **lines 460-481, page 15:**

“Most conventional immunomodulatory agents act by inhibiting activation or reducing proliferation of lymphocytes, notably by targeting purine and pyrimidine biosynthesis. Each of these drugs has its own mechanism of action and is used to manage a variety of autoimmune conditions such as rheumatoid arthritis, psoriasis, systemic lupus erythematosus, systemic sclerosis or organ transplantation. For example, methotrexate inhibit several enzymes responsible for nucleotide synthesis including dihydrofolate reductase, thymidylate synthase, aminoimidazole carboxamide ribonucleotide transformylase (AICART) and, amido-phosphoribosyltransferase⁵⁰. Azathioprine has an antagonist effect on purine metabolism leading to broad inhibition of DNA, RNA, and protein synthesis⁵¹. Another drug commonly used is the mycophenolate, which is an inhibitor of the inosine-5'-monophosphate dehydrogenase, which result in depletion of guanosine nucleotide preferentially in T and B lymphocytes thus inhibiting their proliferation⁵². Teriflunomide selectively and reversibly inhibits dihydro-orotate dehydrogenase, a key mitochondrial enzyme in the de novo pyrimidine synthesis pathway, leading to a reduction in proliferation of activated T and B lymphocytes without causing cell death^{53, 54}. Although these drugs, which interfere with nucleotide synthesis, have been shown to be effective, they have more than one mechanism of action and the precise way in which they exert their effects is often unknown. Moreover, their anti-proliferative and cytotoxic effects are in most cases not specific to the immune system explaining their toxicity and side-effects. In our case, we hypothesized based on our acknowledge of CTPS1, that targeting CTPS1 (with selective inhibitors) would be more specific and could lead to less adverse side-effects.

Indeed, we widely characterized the role of CTPS1 in proliferation in activated T cells and activated T cells represent one of the highest CTPS1 expressing tissue^{11, 12}.”

Q7 The GI issues seen in the mice are not adequately explained and would warrant some transfer studies of normal cell subsets to evaluate if they are protective.

We have now discussed in more details and explained the GI issues to clarify these observations: **lines 161-163, page 6:**

“These observations showed that CTPS1 is key for the proliferation and the renewal of the intestinal epithelium of the gut, which is considered as one of the highly proliferative tissues in the whole body²⁰.”

Q8: Is it possible to do some single cell analysis of the immune systems of the mice to better define the populations affected?

Indeed, single cell RNAseq analyses (and/or CytOF) are interesting approaches for future studies that could help us to better define populations affected by the absence or the inhibition of CTPS1.

Point-by-point response to **Reviewer #3** (expert in pyrimidine metabolism and cell development):

In this work, the authors used mouse models with tissue selective inactivation of CTPS1 and CTPS2, coding for CTP synthetases of the pyrimidine biosynthetic pathway, to elucidate their roles in cell proliferation. They observed that highly proliferative tissues, including activated B and T lymphocytes, and memory T cells, rely heavily on CTPS1. Previous work on the combined immunodeficiency caused by mutation in CTPS1 in humans, led the authors to test the effect of CTPS1 deletion or inhibition on the lethal poly-autoimmunity of Scurfy mice. Their results suggest that CTPS1 should be explored as a potential drug target for treatment of autoimmune and inflammatory diseases. This manuscript represents an extensive amount of work that brings us closer to understanding the role of CTP synthetases in the immune response.

Some minor revisions are required.

INTRODUCTION:

Define GVHD.

We have defined GvHD in our introduction as graft versus host disease. **Lines 100-101, page 3.**

What is the percent identity between human CTPS1, CTPS2, and between the human and mouse enzymes?

Both human CTPS1 and CTPS2 and mouse CTPS1 and CTPS2 share 75% of identity between each other at the protein level. The human CTPS1 protein share 97.8% of identity with the mouse CTPS1 protein, while human CTPS2 protein share 90% of identity with mouse CTPS2 protein.

This information was added **lines 79-81, page 3:**

“These genes share 75% homology/identity^{2, 3}, and are highly conserved in mouse (90% of homology/identity), and locate on chromosomes 1 and X respectively, both in human and mouse.”

Although the authors provide references from the 1970s and 1980s on the role of CTPS in cell proliferation and cancer, additional, more current citations on the state of this research should be provided.

We have added the following recent references on the role of CTPS1 in proliferation and cancer:

-Lin Y, Zhang J, Li Y, Guo W, Chen L, Chen M, Chen X, Zhang W, Jin X, Jiang M, Xiao H, Wang C, Song C, Fu F. CTPS1 promotes malignant progression of triple-negative breast cancer with transcriptional activation by YBX1. *J Transl Med.* 2022 Jan 6;20(1):17. doi: 10.1186/s12967-021-03206-5. PMID: 34991621; PMCID: PMC8734240.

-Minet N, Boschac AC, Lane R, Laughton D, Beer P, Asnagli H, Soudais C, Bourne T, Fischer A, Martin E, Latour S. Differential roles of CTP synthetases CTPS1 and CTPS2 in cell proliferation. *Life Sci Alliance.* 2023 Jun 22;6(9):e202302066. doi: 10.26508/lsa.202302066. PMID: 37348953; PMCID: PMC10288033.

RESULTS:

Provide a possible explanation for why the knock-in of the mutation mimicking the human CTPS1 mutation did not affect CTPS1 expression in the mouse model.

The human mutation, we have reproduced in mouse by CRISPR-Cas9, is affecting a splice site and although homologous-exon sequences are highly conserved in mammalian genomes, intronic counterparts are poorly conserved. For example, there is a low level of conservation of the branch-site sequences in human and mouse genome (Kol G, Lev-Maor G, Ast G. Human-mouse comparative analysis reveals that branch-site plasticity contributes to splicing regulation. *Hum Mol Genet.* 2005 Jun 1;14(11):1559-68. doi: 10.1093/hmg/ddi164. Epub 2005 Apr 27. PMID: 15857856.) Thus, the reasonable explanation for why the knock-in of the mutation mimicking the human CTPS1 mutation did not affect CTPS1 expression in the mouse model is that the splicing constraints for CTPS1 in mouse and human are different and in mice the mutation did not disturb the splicing.

We now added a sentence to explain our hypothesis,

lines 115-117, page 5:

“The possible explanation for this discrepancy is that the splicing constraints for *Ctps1/CTPS1* in mouse and human are different and in mouse the mutation did not disturb the splicing¹⁵.”

A general comment that applies to many figures: the text in the labelling of figures looks out of focus, and is hard to read when in superscripts, see for example Fig. 2 b – e, Fig. 4 b, Fig. 4 d, Fig. 4 f, Fig. 5 a – c, etc.

We apologize and uploaded higher definition for the figures.

Fig. 1a: “Nb” is not a standard abbreviation for “number”.

We apologize for this mistake and change it to Number.

Fig 1d. Why isn't the Ctps1 antibody signal decreased more in the Ctps1 ko/ko embryos? There appears to be substantial signal remaining compared to the Ctps1 wt/flox mice.

Most of the fluorescence intensity reflects specific CTPS1 immunoreactivity, but the fluorescence that remains in Ctps1 ko/ko is probably due to tissue autofluorescence and/or antibody trapping. Immunostaining of whole mount mouse embryos is prone to low level of background staining.

This point is now discussed **lines 135-137, page 5:**

“However, low remaining staining was observed in Ctps1 deficient embryos probably due to tissue autofluorescence and/or antibody trapping as whole mount mouse embryos is prone to low level of background staining.”

Care should be taken to use the same notation in the text and in the figures. The text of the Results, p.5, refers to ERT2, in Supplementary Fig. 2d, the notation is ERT2, and in Supplementary Fig. 3 the notation is ERt2.

We have verified in the literature and standardize our nomenclature to “ER^{T2}” in the text, the figures and supplementary.

Fig. 2a. How many animals are represented?

We added the number of animals represented in the legend of Figure2a.

Fig 2b: In the legend, refer to “black arrowheads” instead of “black sign”.

According to Reviewer#2, we changed to “black arrowhead” in Figure 2b.

Fig. 2 d-f: Correct the legend. The left panels are dot-plots, and the right panels are percentages of B and T cells.

We apologize and have corrected the mistake.

It would be useful to provide the reader with the table shown in Supplementary Fig. 2d in the manuscript, instead of in the supplementary material.

As requested, we moved the table from Supplementary Fig. 2d to Table 1.

Fig. 3d: In the immunoblots, right panel (Activated-CD3/CD28), lanes wt/ko-1, and lane wt/ko-4 look like they have the same intensities of CTPS1 signal as the lane with ko/ko-1. What is the half-life of CTPS1? Would one expect to see residual protein, left over from before the tamoxifen treatment?

Upon T-cell activation CTPS1 is upregulated 24 hours after stimulation and its expression remain till 96 hours. Indeed, the hypothesis raised by reviewer #3 namely a residual expression due to left over from before Tam treatment is very likely.

We have now addressed this point, **lines 201-202, page 7:**

“However, some residual CTPS1 expression was observed in Ctps1^{ko/ko} T cells likely due to incomplete Ctps1 deletion after tamoxifen treatment.”

Fig 7d: In the legend, refer to “yellow arrowheads” instead of “yellow signs”.

According to Reviewer#2, we changed to “black arrowhead” in Figure 7d.

DISCUSSION:

In the discussion the authors should comment that the pyrimidine biosynthetic pathway has

long been considered a target for the treatment of autoimmune disease, although the focus has been on dihydroorotate dehydrogenase (see for example Scherer et al. 2023, PMID: 36797499, doi: 10.1038/s41590-023-01436-x).

As mentioned, the comment from Reviewer#3 echo Reviewer#1 comment. We thus address both comments by adding a paragraph, **lines 460-484, page 15:**

“Most conventional immunomodulatory agents act by inhibiting activation or reducing proliferation of lymphocytes, notably by targeting purine and pyrimidine biosynthesis. Each of these drugs has its own mechanism of action and is used to manage a variety of autoimmune conditions such as rheumatoid arthritis, psoriasis, systemic lupus erythematosus, systemic sclerosis or organ transplantation. For example, methotrexate inhibit several enzymes responsible for nucleotide synthesis including dihydrofolate reductase, thymidylate synthase, aminoimidazole caboxamide ribonucleotide transformylase (AICART) and, amido-phosphoribosyltransferase⁵⁰. Azathioprine has an antagonist effect on purine metabolism leading to broad inhibition of DNA, RNA, and protein synthesis⁵¹. Another drug commonly used is the mycophenolate, which is an inhibitor of the inosine-5'-monophosphate dehydrogenase, which result in depletion of guanosine nucleotide preferentially in T and B lymphocytes thus inhibiting their proliferation⁵². Teriflunomide selectively and reversibly inhibits dihydro-orotate dehydrogenase, a key mitochondrial enzyme in the de novo pyrimidine synthesis pathway, leading to a reduction in proliferation of activated T and B lymphocytes without causing cell death^{53, 54}. Although these drugs, which interfere with nucleotide synthesis, have been shown to be effective, they have more than one mechanism of action and the precise way in which they exert their effects is often unknown. Moreover, their anti-proliferative and cytotoxic effects are in most cases not specific to the immune system explaining their toxicity and side-effects. In our case, we hypothesized based on our acknowledge of CTPS1, that targeting CTPS1 (with selective inhibitors) would be more specific and could lead to less adverse side-effects. Indeed, we widely characterized the role of CTPS1 in proliferation in activated T cells and activated T cells represent one of the highest CTPS1 expressing tissue ^{11, 12}. One possible limitation of the use of CTPS1 inhibitors is to impair the immune response to viral infections including EBV reactivation (as it is observed in CTPS1-deficient patients). This unwanted side effect could be resolved by an adjusted dosage of CTPS1 inhibitor (as it is done for other immunosuppressive drugs).”

Reviewers' Comments:

Reviewer #1:

Remarks to the Author:

The additional discussion is thorough and addresses the points I raised at previous review.

Reviewer #2:

Remarks to the Author:

The authors have answered all of my concerns and I appreciate the extra experiments performed to answer my questions.

Reviewer #3:

Remarks to the Author:

In this revised manuscript, the authors have addressed all the concerns stated in my previous review.

REVIEWERS' COMMENTS

Reviewer #1 (Remarks to the Author):

The additional discussion is thorough and addresses the points I raised at previous review.

Reviewer #2 (Remarks to the Author):

The authors have answered all of my concerns and I appreciate the extra experiments performed to answer my questions.

Reviewer #3 (Remarks to the Author):

In this revised manuscript, the authors have addressed all the concerns stated in my previous review.

We are pleased to learn that the reviewers are satisfied with the changes introduced in the manuscript and with our reviewing responses. Once again, we would like to thank the reviewers for their appreciation of our work, for their comments and experiments proposals which we think, helped us improve the quality of the manuscript.

Yours sincerely,
Claire Soudais and Sylvain Latour